# DIVERSITY MATTERS: REVISITING TEST-TIME COMPUTE IN VISION-LANGUAGE MODELS

## ABSTRACT

Test-time compute (TTC) strategies have emerged as a lightweight approach to boost reasoning in large language models, but their applicability to vision-language models (VLMs) remains unclear. We present a systematic study of TTC for visual reasoning across seven open-source VLMs and six benchmarks, revisiting two paradigms: (i) feature-based scoring of chain-of-thought (CoT) traces and (ii) confidence-based aggregation via majority voting (MV). In the single-model setting, feature cues (e.g., length, pivot words) fail to improve accuracy, while MV yields only modest, CoT-dependent gains. To explain this limitation, we theoretically show that the voting method's effectiveness depends on *prediction diversity*: when outputs are highly correlated, the benefit of voting vanishes. In contrast, *multi-model ensembles* introduce stronger diversity through architectural differences, training data, and scale, making them both more realistic and more promising for TTC. However, MV treats all models equally, leaving it vulnerable to correlated errors from weaker models. To address this, we propose *Entropy-based TTC*, which selects the most confident prediction based on predictive entropy. Our method reduces to MV in the single-model case but, in ensembles, leverages confidence disparities to prioritize stronger models. We prove that our method theoretically outperforms MV under mild dependence assumptions, and empirically show that it consistently surpasses both MV and the best individual model across diverse visual reasoning benchmarks. This demonstrates that smaller models can enhance, rather than hinder, larger ones when combined appropriately, unlocking ensemble gains not achievable with existing TTC strategies.

## 1 INTRODUCTION

Vision-Language Models (VLMs) have recently achieved remarkable performance across a range of visual reasoning benchmarks (Llama Team, 2024; Agrawal et al., 2024; Gemma Team, 2025; Bai et al., 2025; OpenAI, 2023; Gemini Team, 2025). At the same time, the large language modeling (LLM) community has developed a family of *test-time compute* (TTC) strategies, particularly those based on *chain-of-thought* (CoT) prompting, to improve reasoning without modifying model parameters (Snell et al., 2024). These strategies generate multiple outputs per input and then aggregate or rank them to produce more reliable predictions.

In the LLM literature, TTC methods fall broadly into two categories. *Feature-based* methods attempt to estimate the quality of each CoT reasoning trace by analyzing textual signals, such as the presence of specific pivot words (Chang et al., 2025; Lippmann & Yang, 2025), confident linguistic tone (Mao et al., 2025), or the length of the reasoning chain (Fu et al., 2023; Jin et al., 2024). In contrast, *confidence-based* methods treat the model as a stochastic oracle and improve reasoning reliability by aggregating multiple outputs, typically selecting the most frequent answer across samples via voting (Wang et al., 2023; Chen et al., 2024b; Snell et al., 2024).

Applying TTC to VLMs, however, is far from straightforward. Unlike LLMs, VLMs must first perceive and interpret dense visual signals before reasoning over them. This introduces new challenges: (i) visual perception is inherently error-prone and varies across models (Bhattacharyya et al., 2023; Wang et al., 2025); (ii) vision-language alignment remains imperfect, creating subtle inconsistencies (Li et al., 2025; Yan et al., 2025); and (iii) textual cues that correlate with the correctness in LLM

may not reflect the true visual understanding (Al-Tahan et al., 2024; Jiang et al., 2025). Therefore, it is unclear whether and when TTC strategies can reliably enhance visual reasoning.

To investigate this, we begin with the *single-model (multi-round)* setting, where one VLM is queried multiple times with randomness (§ 3). Our findings reveal that: (1) feature-based methods fail to improve accuracy, showing that linguistic style is a poor proxy for visual reasoning quality; and (2) confidence-based methods such as majority voting (MV) provide only modest, but consistent, gains, and only when CoT prompting is used. Without CoT, even aggregation brings no benefit.

Why are these gains so limited? We analyze the *diversity* (formally, the *statistical dependency*) between predictions and show that MV's effectiveness decreases as predictions become more correlated (§ 4.1). When model outputs are nearly identical, voting cannot amplify the signal of correctness. Empirically, we confirm this across 7 VLMs and 6 datasets: outputs exhibit weak but nonzero dependency, which explains why MV offers only small improvements in practice (§ 4.2).

These insights point to a deeper limitation: in the single-model setting, diversity arises only from sampling randomness, so the expected skill of the model remains unchanged. By contrast, *multi-model ensembles* naturally introduce stronger diversity: differences in architecture, training data, and even scale create complementary strengths. This makes ensembles both more realistic in practice and more promising for TTC. Existing methods, such as MV, cannot exploit this potential: by treating all models equally, MV risks letting weaker but correlated models dominate the outcome. What is needed is a strategy that adapts to model quality and selectively prioritizes the most reliable predictions.

To address this, we introduce a new TTC strategy for visual reasoning: *Entropy-based Test-Time Consistency (ETTC)* (§ 5.1). Instead of counting votes, ETTC selects the prediction with the lowest entropy (on the answer distribution from multiple responses), that is, the most confident output distribution. In the single-model setting, ETTC reduces to MV, ensuring backward compatibility. But in multi-model ensembles, ETTC diverges from MV: it leverages confidence gaps across models, allowing smaller models to assist stronger ones rather than overwhelm them. We theoretically prove that ETTC outperforms MV under mild dependence assumptions (§ 5.2), and empirically show that it not only improves over MV but can even surpass the best individual model in the ensemble (§ 5.3). This result is particularly striking: *smaller models can be used to enhance larger ones when combined wisely*, yielding gains not achievable with MV alone.

In summary, our contributions are:

- A systematic theoretical and empirical study of TTC in VLMs, showing that feature cues fail and that MV yields only modest CoT-dependent gains (§ 3).

- A theoretical analysis linking MV's effectiveness to prediction dependency, supported by empirical evidence across diverse models and datasets (§ 4).

- A new entropy-based method, ETTC, that generalizes MV and achieves consistent improvements in multi-model ensembles, often surpassing even the best single model (§ 5).

## 2 PREPARATION

We begin by outlining the models, datasets, prompting formats, TTC baselines, and general evaluation settings used in our experiments.

**Models.** We evaluate seven open-source VLMs under two complementary multi-model ensemble configurations. In the *similar-size (cross-family)* setup, we include four VLMs with comparable parameter sizes but diverse architectures: Qwen2.5-VL-7B-Instruct (Bai et al., 2025, Qwen-7B), LLaMA-3.2-11B-Vision (Llama Team, 2024, LLaMA), Gemma-3-12B-it (Gemma Team, 2025, Gemma), and Pixtral-12B-2409 (Agrawal et al., 2024, Pixtral). In the *same-family (varied-size)* setup, we use four models from the Qwen2.5-VL-Instruct family (Bai et al., 2025), ranging from 3B to 72B parameters (3B, 7B, 32B, 72B), allowing us to study scaling effects within a single model family.

**Datasets.** We experiment on six multiple-choice visual QA benchmarks covering three domains. For *math reasoning*, we use the testmini split of MathVista (Lu et al., 2024) and the test set of MathVision (Wang et al., 2024). For *diagram understanding*, we include the test sets of TQA (Kim et al., 2019) and ScienceQA (Lu et al., 2022). For *general visual reasoning*, we use the validation

splits of MMStar (Chen et al., 2024a) and MMMU (Yue et al., 2024). All datasets contain multiple-choice QA instances with $K$ answer options per question ($2 \leq K \leq 9$). Further statistics, including domain, split size, and option counts, are summarized in Tab. 3 in App. C.1.

**Decoding.** We use decoding (Sutskever et al., 2014) via HuggingFace's default generation settings.[1] We adopt two prompting formats: (1) *Non-CoT (n-CoT)* prompting discourages intermediate reasoning and elicits direct answers; (2) *Chain-of-thought (CoT)* prompting explicitly encourages step-by-step reasoning, followed by a final answer. We use zero-shot, one-stage prompting for both settings to ensure consistency across models. Full prompt templates are provided in Figs. 4 and 5 in App. C.2. Final answers are parsed via regex to extract discrete predictions.

**TTC baselines.** To revisit TTC strategies for visual reasoning, we evaluate four representative baselines spanning feature-based and confidence-based approaches. Three are *feature-based* scoring methods applied to CoT responses: (1) *CoT Pivot Word* ranks each response by counting predefined reasoning-related expressions (e.g., "alternatively") (Chang et al., 2025; Lippmann & Yang, 2025); see full phrase list in Tab. 4 of App. C.3. (2) *CoT Length* prefers longer responses, following prior work suggesting a correlation between length and reasoning quality (Fu et al., 2023). (3) *Feature-All* combines four interpretable features—pivot word count, vague word count, total token count, and lexical diversity—to compute a composite score (see Tab. 6). As a *confidence-based* method, (4) *Majority Voting (MV)* (Wang et al., 2023; Snell et al., 2024) aggregates $N = 16$ samples and selects the most frequent final answer (breaking ties randomly).

**Evaluation settings.** We assess all TTC methods under two settings: (1) In the *single-model (multi-round)* setting, a single VLM is queried $N$ times per question with stochasticity in decoding (e.g., CoT sampling). TTC is used to aggregate these intra-model outputs. (2) In the *multi-model ensemble* setting, $M$ distinct VLMs are queried per question (each with multiple samples), introducing both intra- and inter-model variation. This setting allows us to study cross-model complementarity and test whether aggregating weaker models can improve over any individual model.

## 3 WHETHER TTC WORKS IN VISUAL REASONING

We begin by revisiting whether TTC strategies, widely used in LLMs, improve visual reasoning in VLMs. We evaluate four representative methods across six multiple-choice visual benchmarks and compare their performance under two prompting conditions: direct answering (n-CoT) and chain-of-thought reasoning (CoT). Results are averaged across seven VLMs unless otherwise noted.

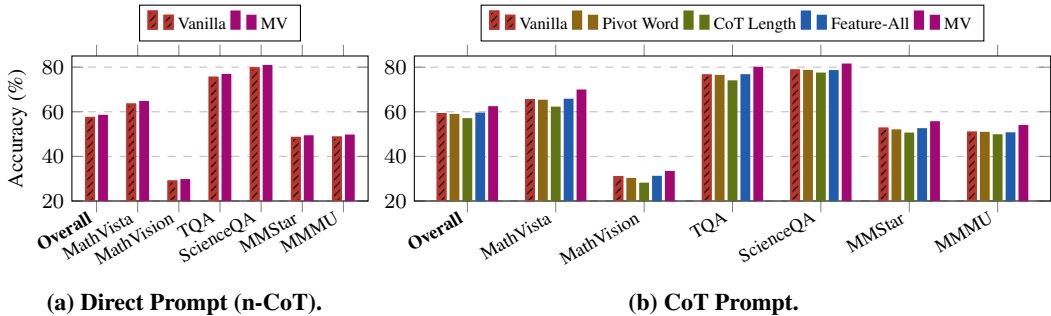

(a) Direct Prompt (n-CoT).  (b) CoT Prompt.

Figure 1: Comparison of test-time compute (TTC) strategies under two prompting styles. In **n-CoT** (left), models are instructed to output only the final answer without reasoning; feature-based methods are inapplicable, and majority voting (MV) shows no improvement. In **CoT** (right), models are prompted to reason step by step. While feature-based methods yield no gains, MV offers modest but consistent improvement across datasets.

**Direct Prompt (n-CoT): TTC fails without CoT.** The *n-CoT* setting tests whether test-time variation alone, without prompting explicit reasoning, can boost accuracy. Since no reasoning chains are produced, only confidence-based methods like majority voting (MV) are applicable.

As shown in Fig. 1 (left), MV provides negligible or no improvement over the greedy baseline (often <1%). Although we sample 16 outputs per question with stochastic decoding, the model's predictions

---

[1]https://huggingface.co/docs/transformers/en/generation_strategies

are mostly identical. This suggests that in the absence of CoT prompting, VLMs tend to output the same surface-level answer, showing little diversity in reasoning or interpretation. As a result, TTC offers no benefit under direct answering. This aligns with findings in LLMs (Wang et al., 2023; Snell et al., 2024), but is further exacerbated in VLMs due to the perception bottleneck, visual content must first be interpreted before any meaningful variation can emerge.

**Chain-of-Thought Prompt (CoT): confidence helps, features don't.** In contrast, when models are prompted to reason step-by-step using CoT, test-time strategies have room to work. This setup enables both feature-based (e.g., CoT length, pivot words) and confidence-based (e.g., MV) approaches.

As shown in Fig. 1 (right), MV consistently improves performance across all benchmarks, with average gains of 2-4%. This validates the utility of test-time sampling under CoT: the model explores diverse reasoning paths and occasionally corrects itself. However, the improvements are modest, suggesting that sampled CoTs are still highly correlated, a hypothesis we will formally investigate in § 4. Meanwhile, feature-based methods fail to provide any consistent gain over vanilla CoT. Their performance often fluctuates slightly around the baseline. This highlights a key difference from LLMs: in VLMs, textual heuristics are poor proxies for reasoning correctness because visual understanding is the bottleneck. If perception fails, even a well-formed CoT cannot save the answer.

**Takeaway.** TTC can improve visual reasoning, but only under specific conditions. Without CoT prompting, models produce nearly identical outputs, leaving no room for improvement. Even with CoT, gains from MV are modest, and feature-based scoring fails to help, highlighting the unique challenges of visual reasoning where perception quality limits downstream reasoning. This raises a key question: *when does TTC actually help?* To answer this, we now turn to the analysis of MV, focusing on how its effectiveness depends on the statistical dependencies among model predictions.

## 4    WHEN DOES TTC WORK IN VISUAL REASONING?

Why does test-time compute (TTC), especially majority voting (MV), sometimes fail to improve accuracy in visual reasoning? We address this question by analyzing how the statistical dependency among model predictions influences the effectiveness of MV. To this end, we develop a theoretical framework that quantifies this relationship and support it with empirical evidence.

### 4.1    THEORETICAL INSIGHT: TTC HELPS WHEN PREDICTIONS ARE DIVERSE

**Setup.** Consider a $K$-choice question with a unique correct answer $Y \in [K]$. Let $X_1, \ldots, X_U \in [K]$ be $U$ predictions, either from $U$ decoding samples of a single VLM or from $U$ different VLMs in an ensemble.[2] Define the correctness indicator $Z_u := \mathbb{I}\{X_u = Y\}$ and let the single-trial accuracy be $p := \mathbb{E}[Z_u]$. Let $S_k := \sum_{u=1}^{U} \mathbb{I}\{X_u = k\}$ denote the number of votes for option $k$, and let the MV prediction be $\widehat{Y}_{\mathrm{MV}} := \arg\max_k S_k$. Define the MV accuracy as $A_{\mathrm{MV}}(U) := \mathbb{P}(\widehat{Y}_{\mathrm{MV}} = Y)$, and the improvement as $\Delta A_{\mathrm{MV}}(U) := A_{\mathrm{MV}}(U) - p$.

**Dependency metrics.** To understand when MV is effective, we quantify the *dependency* among predictions using two metrics: *normalized mutual information (NMI)* and *correlation*. For answer variables $X, X'$, we define NMI as

$$\mathrm{NMI}(X; X') := \frac{I(X; X')}{\min\{H(X), H(X')\}}, \quad H(X) = -\sum_{k=1}^{K} \mathbb{P}(X = k) \log \mathbb{P}(X = k).$$

For $U$ predictions, the average NMI is:

$$\overline{\mathrm{NMI}} := \frac{2}{U(U-1)} \sum_{u < v} \mathrm{NMI}(X_u; X_v).$$

For correctness indicators $Z, Z'$, define the *correlation* as

$$\rho(Z, Z') := \frac{\mathbb{E}[ZZ'] - p^2}{p(1-p)}, \quad \overline{\rho} := \frac{2}{U(U-1)} \sum_{u < v} \rho(Z_u, Z_v).$$

---

[2]The theoretical result holds regardless of the origin of the $U$ predictions.

**Theorem 1.** *Suppose all prediction pairs $(X_u, X_v)$ share the same dependency level (i.e., $\overline{\mathrm{NMI}}$ or $\overline{\rho}$). Then the MV improvement $\Delta A_{\mathrm{MV}}(U)$ is monotonically decreasing in both $\overline{\rho}$ and $\overline{\mathrm{NMI}}$. In particular:*

$$\overline{\rho} = 1 \ (or \ \overline{\mathrm{NMI}} = 1) \Rightarrow \ \Delta A_{\mathrm{MV}}(U) = 0,$$

$$\overline{\rho} = 0 \ (or \ \overline{\mathrm{NMI}} = 0), \ p > \tfrac{1}{K} \Rightarrow \ A_{\mathrm{MV}}(U) \to 1 \ as \ U \to \infty.$$

**Interpretation.** The proof is provided in App. B.1. This theorem reveals a simple but powerful insight: *MV only improves accuracy when predictions are diverse.* If all predictions are identical (i.e., fully dependent), MV reduces to a single prediction, yielding no gain. But if predictions are uncorrelated and individually better than random guessing ($p > 1/K$), MV can aggregate signal and achieve near-perfect accuracy as the number of predictions $U$ grows. Both $\overline{\rho}$ and $\overline{\mathrm{NMI}}$ are practical, interpretable, and model-agnostic indicators of this diversity. Thus, they can serve as useful tools to estimate when TTC is likely to help, without relying on ground truth labels or model internals.

### 4.2 EMPIRICAL VERIFICATION

We now provide empirical evidence to support our theoretical findings in § 4.1. In particular, we examine how model prediction dependency, quantified by $\overline{\mathrm{NMI}}$ and $\overline{\rho}$, affects MV performance. Our goal is twofold: (1) determine how many decoding samples $U$ are sufficient to obtain stable dependency estimates and maximal MV improvement, and (2) empirically verify the theoretical prediction that MV improvement decreases with increasing dependency.

#### 4.2.1 HOW MANY DECODING SAMPLES ARE SUFFICIENT?

Our theoretical analysis assumes a sufficiently large number of decoding samples $U$, such that MV benefits fully materialize. In practice, however, increasing $U$ incurs additional computational cost. Thus, we first investigate the convergence of dependency metrics as $U$ grows, aiming to find the minimal $U$ that yields stable estimates.

**Setup.** We use Qwen-7B to generate $U = 2$ to 16 decoded outputs for each example across six visual reasoning datasets. For each $U$, we compute two dependency metrics: average normalized mutual information $\overline{\mathrm{NMI}}$ and average correctness correlation $\overline{\rho}$ between response pairs.

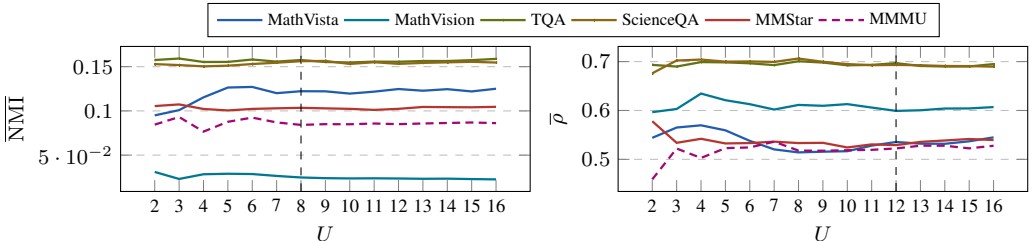

Figure 2: Convergence of dependency with decoding sample size $U$ on Qwen-7B. Both $\overline{\mathrm{NMI}}$ and $\overline{\rho}$ stabilize when $U{=}12$, suggesting that a moderate number of samples is sufficient to estimate dependency reliably.

**Findings.** As shown in Fig. 2, both $\overline{\mathrm{NMI}}$ and $\overline{\rho}$ stabilize around $U = 12$ across all datasets. Beyond this point, additional samples offer minimal benefit in estimating prediction dependency. Sampling more than 12 responses provides diminishing returns in estimating dependency. Thus, we use $U = 16$ in all subsequent experiments to ensure both stability and tractability.

#### 4.2.2 DOES MV IMPROVEMENT DECREASE WITH DEPENDENCY?

Next, we test our core theoretical prediction: MV is most beneficial when model outputs are diverse. That is, MV improvement should decrease as prediction dependency increases.

**Setup.** We evaluate MV improvement $\Delta A_{\mathrm{MV}}(16)$ for seven models across six datasets, using $U = 16$ decoding samples. For each model, we compute the average improvement and average dependency across datasets, measuring dependency with both $\overline{\mathrm{NMI}}$ and $\overline{\rho}$.

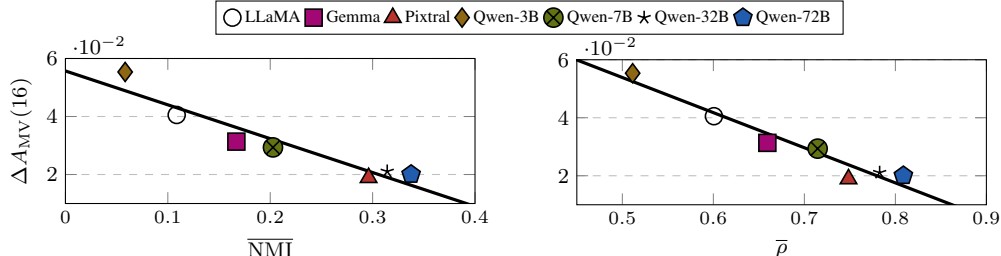

Figure 3: MV improvement decreases with higher prediction dependency. Across models, MV improvement $\Delta A_{\mathrm{MV}}(16)$ is negatively correlated with both $\overline{\mathrm{NMI}}$ and $\bar{\rho}$, confirming theoretical predictions.

**Findings.** Fig. 3 shows a clear negative correlation between MV improvement and both dependency metrics. Smaller models (e.g., Qwen-3B, LLaMA), which produce more diverse outputs, benefit more from MV. In contrast, larger or more deterministic models (e.g., Qwen-72B, Pixtral) exhibit limited diversity and gain less from MV. Detailed results are in Figs. 6 and 7 in App. D.1.

**Takeaway.** MV effectiveness hinges on the diversity of model outputs. As predictions become more deterministic, reflected by higher dependency metrics such as $\overline{\mathrm{NMI}}$ and $\bar{\rho}$, MV offers diminishing returns. This empirical trend aligns with our theoretical findings and suggests a practical principle: MV is most beneficial when applied to weaker or smaller models, or in settings where the model's confidence is low and outputs are more stochastic. For example, in few-shot or domain-shifted scenarios where models are uncertain, decoding diversity tends to be higher, allowing MV to amplify weak but complementary signals. Conversely, when using large, over-optimized models that produce highly consistent predictions (e.g., Qwen-72B), MV is unlikely to help and may introduce unnecessary compute cost. Overall, this analysis provides a practical lens for when and why to apply TTC strategies like MV in real-world visual reasoning tasks.

## 5 BEYOND MV: ENTROPY-BASED TTC FOR MULTI-MODEL ENSEMBLES

Building on the insight that MV benefits from diverse yet independent predictions, we now turn to the more realistic and underexplored *multi-model ensemble* setting. Compared to multi-round decoding from a single model, where prediction diversity is limited, ensembles of heterogeneous models naturally offer complementary strengths and errors. Here, we first introduce an entropy-based TTC method (ETTC) designed to better leverage cross-model diversity. We then theoretically show that ETTC outperforms MV under mild conditions, and empirically demonstrate that it enables smaller models to enhance or even surpass larger ones in visual reasoning.

### 5.1 ENTROPY-BASED TTC (ETTC)

Our previous analysis showed that the effectiveness of MV depends heavily on prediction diversity. However, MV has a deeper limitation in multi-model ensemble settings: it assumes all model responses are equally reliable and votes based solely on frequency, ignoring how confident or capable each model is. This oversight is less problematic in the single-model setting, since all predictions come from the same model, their expected quality is the same. But in multi-model ensembles, where models vary in size, training, and performance, this uniform treatment becomes a liability. A majority of weaker models can outvote a stronger one, even when the latter is confidently correct.

To address this, we introduce Entropy-Based Test-Time Compute (ETTC): a simple, model-agnostic method that selects the most confident prediction among multiple sources, rather than relying on vote counts. ETTC uses normalized predictive entropy as a proxy for confidence.

**Definition 1** (Entropy-Based Selection Rule). *Let $U$ sources (models or decoding rounds) each produce a predictive distribution $p_u(\cdot) \in \Delta^{K-1}$ over $K$ answer options. Define the normalized entropy as*

$$\widetilde{H}_u := -\frac{1}{\log K} \sum_{k=1}^{K} p_u(k) \log p_u(k) \in [0, 1],$$

*and the top-1 prediction $\hat{y}_u := \arg\max_k p_u(k)$. ETTC selects the least-uncertain source,*

$$u^\star := \arg\min_{u \in [U]} \widetilde{H}_u, \qquad \widehat{Y}_{\min H} := \hat{y}_{u^\star}.$$

This selection rule prioritizes predictions with lower uncertainty, under the intuition that higher model confidence often correlates with correctness, especially for well-calibrated or stronger models. In contrast to MV, which can amplify weak or erroneous signals through majority effects, ETTC amplifies precision by trusting the most decisive prediction. Notably, ETTC reduces to MV in the single-model multi-round case when we average predictive distributions and pick the most probable option. But in the multi-model setting, it diverges: it allows stronger models to dominate the decision, even when they are in the minority, an essential property for leveraging heterogeneous ensembles.

**Takeaway.** ETTC replaces vote count with model confidence, providing a more principled and adaptive aggregation strategy for ensemble reasoning. Especially in real-world scenarios where model capabilities vary, ETTC is better equipped to avoid over-reliance on weaker models and better exploit the reliability of stronger ones.

## 5.2 THEORETICAL INSIGHT: ETTC OUTPERFORMS MV IN MULTI-MODEL ENSEMBLES

In the multi-model ensemble setting, models vary in strength and reliability, which increases the answer diversity. While MV treats models equally, this can backfire: weaker models may collectively outvote stronger ones, especially when their errors are correlated. Our goal is to theoretically understand why the proposed ETTC method provides a more robust alternative in such scenarios.

We begin by formalizing a key intuition: *more confident predictions tend to be more accurate*.

**Assumption 1** (Entropy-Accuracy Monotonicity)**.** *For a given input with true label $Y$, suppose model $u$ assigns probability $p_u(Y)$ to $Y$, and $\widetilde{H}_u$ is its normalized entropy. Then, for all $u, v \in [U]$:*

$$p_u(Y) > p_v(Y) \quad \Rightarrow \quad \widetilde{H}_u < \widetilde{H}_v.$$

This assumption states that a model assigning a higher probability to the correct answer also tends to be more confident (i.e., has lower entropy). While this relationship may not hold perfectly, we find that it holds approximately in practice across datasets and models (see Fig. 8 in App. D.2).

Given this, ETTC simply selects the prediction from the most confident model (i.e., with lowest entropy on the answer distribution). Let $c^\star := \Pr(\hat{y}_{u^\star} = Y)$ be the accuracy of the most accurate model $u^\star$. ETTC guarantees performance at least $c^\star$, and may occasionally do better by selecting another model whose prediction is both confident and correct. To model dependency among models, we consider a simple coupling scheme: with probability $\lambda$, all non-best models copy the same prediction $W$ (e.g., due to shared biases or training data); with probability $1 - \lambda$, their predictions are conditionally independent. Let $\bar{c} := \Pr(W = Y)$ be the accuracy of this "bloc" prediction, and $A_{\mathrm{MV}}(0)$ be the MV accuracy in the fully independent case.

**Theorem 2** (Superiority of ETTC over MV)**.** *With the setup above and under Assumption 1, let $A_{\min H} := \Pr(\hat{y}_{\min H} = Y)$ be the ETTC accuracy. Then for all $\lambda \in [0, 1]$, we have:*

$$A_{\mathrm{MV}}(\lambda) = \lambda\,\bar{c} \;+\; (1 - \lambda)\,A_{\mathrm{MV}}(0), \tag{1}$$

$$A_{\min H} - A_{\mathrm{MV}}(\lambda) = \lambda(c^\star - \bar{c}) \;+\; (1 - \lambda)(A_{\min H} - A_{\mathrm{MV}}(0)). \tag{2}$$

*In particular, $A_{\min H} \geq A_{\mathrm{MV}}(\lambda)$ for all $\lambda$, with strict inequality whenever $\lambda > 0$ and $\bar{c} < c^\star$.*

**Interpretation.** The proof is in App. B.2. This result highlights a fundamental difference between ETTC and MV in multi-model ensembles. MV aggregates predictions without considering model quality, making it vulnerable to correlated errors, especially when several weaker models dominate the vote. As the error correlation increases (i.e., higher $\lambda$), MV accuracy degrades and converges to that of the bloc prediction $\bar{c}$, which may be substantially lower than the best model's accuracy $c^\star$. In contrast, ETTC avoids this failure mode by selecting the most confident prediction. Under a mild assumption that lower entropy correlates with higher accuracy, ETTC guarantees performance at least as good as the most accurate model, and can even exceed it in practice. Since VLMs often share training data or architecture, making their predictions dependent, ETTC offers a more robust and principled strategy for test-time inference in ensemble settings.

## 5.3 EMPIRICAL VERIFICATION

We now evaluate ETTC in practical multi-model ensemble settings and compare its performance to MV. While our theory highlights ETTC's robustness under dependency, here we empirically verify its effectiveness across two representative ensemble configurations: (1) diverse models of similar size from different families, and (2) scaled models within the same architecture family.

### 5.3.1 SIMILAR-SIZED MODELS FROM DIFFERENT FAMILIES

This experiment evaluates whether ETTC can better leverage diversity among models of comparable size but distinct families. In this setting, models differ in architecture, training data, and accuracy, offering complementary strengths, but also potential variance in prediction quality and confidence.

**Setup.** We select four models of similar scale (7B-12B): LLaMA, Pixtral, Gemma, and Qwen-7B. These models produce predictions for each dataset, and we compare MV and ETTC on the same set of outputs. Notably, no single model consistently dominates across all tasks, and some (e.g., LLaMA) are clearly weaker, adding noise to aggregation.

Table 1: Comparison of ETTC and MV in the multi-model ensemble setting with *similar-sized models from different families*. ETTC consistently outperforms MV across all six datasets, with particularly large gains on benchmarks where model accuracies vary widely (e.g., MathVista, MathVision). This highlights ETTC's ability to prioritize stronger models when aggregating predictions.

| Accuracy (%) | Models | | | | Average | MV | ETTC |
|---|---|---|---|---|---|---|---|
| | LLaMA | Pixtral | Gemma | Qwen-7B | | | |
| MathVista | 52.04 | 56.03 | 65.03 | 72.08 | 61.30 | 68.33 | **75.93** |
| MathVision | 23.41 | 25.20 | 31.84 | 30.18 | 27.66 | 32.05 | **35.57** |
| TQA | 70.41 | 77.34 | 78.86 | 78.50 | 76.28 | 83.65 | **83.90** |
| ScienceQA | 77.84 | 78.32 | 77.83 | 79.76 | 78.44 | **85.52** | 85.28 |
| MMStar | 46.09 | 50.35 | 53.40 | 56.77 | 51.65 | 59.27 | **60.07** |
| MMMU | 42.87 | 47.65 | 52.49 | 50.53 | 48.39 | 53.66 | **58.63** |
| **Average** | 52.11 | 55.82 | 59.91 | 61.30 | 57.29 | 63.75 | **66.56** |

**Findings.** As shown in Tab. 1, ETTC outperforms MV on five of six datasets, with an average accuracy gain of +2.81% (66.56% vs. 63.75%). Larger improvements are seen on tasks where model performance diverges significantly, such as MathVista and MathVision. In these cases, MV suffers from equal-weighting all predictions, allowing weaker models to dilute the ensemble's signal. In contrast, ETTC adaptively prioritizes high-confidence predictions, often aligning with the stronger model per item, and in some cases even exceeding the best model's standalone performance.

**Takeaway.** When aggregating diverse but uneven models, ETTC offers a clear advantage: it selectively filters noise from weaker models based on confidence, making it particularly effective in heterogeneous ensemble settings where voting can be misled by inaccurate predictions.

### 5.3.2 SAME-FAMILY MODELS OF DIFFERENT SCALES

This experiment examines whether ETTC remains effective when models share the same architecture and training data, but differ in scale. While such ensembles may suffer from prediction correlation due to shared inductive biases, scaling laws suggest that performance gaps between model sizes can still introduce meaningful diversity into their predictions.

**Setup.** We use four models from the Qwen family: 3B, 7B, 32B, and 72B. Each model produces predictions on all datasets, and we compare MV and ETTC on their combined outputs. Since all models come from the same training pipeline, this setting represents a high-dependency ensemble, posing a challenge for MV. However, scaling-induced performance gaps can create asymmetric confidence signals that ETTC may exploit.

Table 2: Comparison of ETTC and MV in the multi-model ensemble setting using *same-family models* (Qwen) of increasing scale. ETTC consistently outperforms MV across all datasets, even under highly correlated predictions. Gains are especially pronounced when model accuracies increase with scale, demonstrating ETTC's advantage in prioritizing stronger models within homogeneous ensembles.

| Accuracy (%) | Models | | | | Average | MV | ETTC |
|---|---|---|---|---|---|---|---|
| | Qwen-3B | Qwen-7B | Qwen-32B | Qwen-72B | | | |
| MathVista | 51.94 | 72.08 | 78.58 | 80.58 | 70.80 | 83.15 | **84.44** |
| MathVision | 22.27 | 30.18 | 38.80 | 42.89 | 33.53 | 41.32 | **44.84** |
| TQA | 60.85 | 78.50 | 83.06 | 84.52 | 76.73 | 84.90 | **86.70** |
| ScienceQA | 66.67 | 79.76 | 84.21 | 84.64 | 78.82 | 84.04 | **85.03** |
| MMStar | 41.22 | 56.77 | 56.34 | 62.56 | 54.22 | 61.00 | **63.73** |
| MMMU | 37.41 | 50.53 | 59.04 | 64.18 | 52.79 | 58.63 | **65.34** |
| Average | 46.73 | 61.30 | 66.67 | 69.90 | 61.15 | 68.84 | **71.68** |

**Findings** As shown in Tab. 2, ETTC outperforms MV on all datasets, achieving an average gain of +2.84% (71.68% vs. 68.84%). While overall prediction correlation is higher than in the cross-family setting, the performance variance introduced by scale still provides useful diversity, particularly when smaller models make correct predictions with higher certainty than their larger counterparts. ETTC is able to detect and leverage these instances, occasionally selecting smaller models to override incorrect large-model predictions. In general, ETTC surpasses the accuracy of the strongest model (Qwen-72B) while MV sometimes provides worse performance compared to the strongest model. This show the ability of ETTC to dynamically integrate strengths across the scale spectrum.

**Takeaway.** Despite architectural homogeneity, ensembles of different-sized models still benefit from confidence-based selection. ETTC not only avoids overcounting correlated errors but also allows smaller models to meaningfully enhance or correct the outputs of larger ones, challenging the conventional wisdom that bigger models alone should dominate in test-time ensembles. This highlights ETTC's potential as a lightweight, plug-and-play strategy for amplifying large model performance with smaller, cheaper components.

**Overall Summary.** Across both ensemble settings, diverse and redundant, ETTC consistently outperforms MV without requiring additional training or tuning. These results empirically confirm our theoretical findings: when dependency undermines voting, entropy-based selection offers a more robust and adaptive path to test-time improvement in visual reasoning tasks.

**Supervised Variant of ETTC.** We further extend ETTC to a supervised variant that learns to calibrate confidence signals based on past correctness (App. D.3). We show that even a lightweight classifier trained with minimal supervision significantly improves performance over (unsupervised) ETTC. This suggests that combining confidence with supervised trust modeling offers a promising direction for more adaptive test-time strategies.

## 6 CONCLUSION

We present a comprehensive study of test-time compute (TTC) strategies for visual reasoning, focusing on when and how repeated inference can improve accuracy. Our theoretical and empirical analyses reveal that the effectiveness of majority voting (MV) is tightly linked to the diversity and independence of predictions. While MV offers gains in low-dependency regimes, it fails when outputs are correlated or dominated by weak models. To address these limitations, we propose ETTC: an entropy-based method that selects the most confident prediction, along with a supervised variant that learns when low-entropy signals are reliable. Both methods consistently outperform MV across settings, enabling smaller models to boost larger ones in multi-model ensembles. Our findings highlight confidence, not frequency, as the key to robust TTC in visual reasoning, and offer simple, scalable methods for improving performance without retraining or fine-tuning.

ETHICS STATEMENT

This work does not involve human subjects, sensitive data, or potentially harmful applications. All datasets used are publicly available and widely adopted in the vision-language and reasoning communities. We follow best practices in data handling, model evaluation, and reproducibility, and adhere to the ICLR Code of Ethics in all aspects of our research.

REPRODUCIBILITY STATEMENT

We provide all necessary details to ensure the reproducibility of our work. Model descriptions, experimental setups, and theoretical assumptions are described in the main text and appendix. Complete proofs of theoretical results are provided in App. B. Code and evaluation scripts will be released publicly upon publication.

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

## A    RELATED WORK

**Test-time compute and chain-of-thought in LLMs.**    Chain-of-thought (CoT) prompting improves multi-step reasoning in large language models (Wei et al., 2022; Kojima et al., 2022), and *self-consistency* further boosts accuracy by sampling diverse reasoning paths and selecting the most consistent answer (Wang et al., 2023). Recent work studies how to allocate *test-time compute* (TTC) adaptively and optimally across inputs, showing that compute-optimal scaling of inference-time strategies can rival or exceed scaling model size (Snell et al., 2024). These ideas motivate our transfer of TTC from text-only LMs to VLMs.

**Test-time compute for VLMs and multimodal CoT.**    CoT has been adapted to multimodal reasoning and VQA, including *visual chain-of-thought* prompts and iterative "see-think-confirm" procedures (Chen et al., 2024c). Emerging work explores *test-time consistency* objectives or prompt/ensemble strategies for VLMs, indicating that inference-time aggregation can improve semantic and answer-level consistency without retraining (Chou et al., 2025; Movva & Marupaka, 2025). Our study provides a systematic examination focused on visual multiple-choice reasoning and shows when TTC helps via dependency analysis.

**Ensembling, uncertainty, and correlation.**    Classic results link ensemble gains to *diversity* (low error correlation) among members (Tumer & Ghosh, 1996; Kuncheva & Whitaker, 2003). Deep ensembles effectively capture predictive uncertainty (Lakshminarayanan et al., 2017) and confidence calibration remains critical when aggregating predictions (Guo et al., 2017). From a probabilistic aggregation perspective, our entropy-based selection relates to confidence-weighted "opinion pooling" (Rufo & Pérez, 2012; Dietrich & List, 2017), but we operate at test time with *per-item* uncertainty to decide which model to trust, rather than pooling full distributions.

**Visual reasoning benchmarks and evaluation.**    We evaluate on diverse visual reasoning datasets spanning math, science/diagram, and general multimodal competence: MathVista (visual math reasoning) (Lu et al., 2024), ScienceQA (multimodal science QA with explanations) (Lu et al., 2022), MMMU (college-level multi-discipline reasoning) (Yue et al., 2024), and MMStar (vision-indispensable evaluation) (Chen et al., 2024a). These benchmarks stress perception *and* reasoning, making them suitable for analyzing when TTC helps.

**Reinforcement learning for multimodal reasoning.**    Post-training with RL/RLHF has been explored to improve multimodal alignment and reasoning (Sun et al., 2024; Yu et al., 2024). Such approaches typically require substantial labeled or preference data and non-trivial training budgets. In contrast, our method is a *test-time* procedure; a lightweight supervised variant needs only a small labeled set (e.g., 128 examples) for calibration.

## B    THEORETICAL PROOFS

### B.1    PROOF OF THEOREM 1

*Proof.* We provide a theoretical justification for the claim that the improvement from majority voting (MV) decreases monotonically with statistical dependency among model predictions. We proceed by defining a simple probabilistic coupling model that controls prediction dependency, and then analyze how the expected MV accuracy varies with this dependency level.

#### B.1.1    COUPLING MODEL: COPY-OR-INDEPENDENT SAMPLING

We assume all $U$ predictions $\{X_u\}_{u=1}^{U}$ are drawn from a shared coupling mechanism that depends on a parameter $\lambda \in [0, 1]$: With probability $\lambda$, all predictions are identical copies of a single sample $X$. With probability $1 - \lambda$, predictions are sampled independently from a shared categorical distribution $\pi = (\pi_1, \ldots, \pi_K)$ over $K$ options. Formally, for any pair $(X_u, X_v)$,

$$(X_u, X_v) \sim \begin{cases} (X, X), & \text{with probability } \lambda \\ (X', X''), \quad X', X'' \overset{\text{i.i.d.}}{\sim} \pi, & \text{with probability } 1 - \lambda \end{cases} \tag{1}$$

This ensures uniform pairwise dependency, controlled by $\lambda$.

### B.1.2 LEMMA: BEHAVIOR OF DEPENDENCY METRICS UNDER COUPLING

We now show that both statistical dependency metrics used in our main theorem, normalized mutual information and correctness correlation, are monotonic in $\lambda$ under this coupling.

**(a) Normalized Mutual Information.** Let $X, X'$ be two predictions drawn according to the coupling in equation 1. Their joint distribution is

$$P_\lambda(i, j) = \lambda \cdot \pi_i \cdot \delta_{ij} + (1 - \lambda) \cdot \pi_i \cdot \pi_j,$$

where $\delta_{ij}$ is the Kronecker delta. The marginal distributions remain unchanged as $\pi$.

Since mutual information $I(X; X')$ increases with $\lambda$ (via the convexity of KL divergence), and the marginals are fixed, the normalized mutual information $\mathrm{NMI}(X; X')$ is also non-decreasing in $\lambda$:

$$\mathrm{NMI}(X; X') = \frac{I(X; X')}{H(X)} \uparrow \text{ as } \lambda \uparrow.$$

Hence, the average pairwise NMI $\overline{\mathrm{NMI}}$ is also monotonic in $\lambda$.

**(b) Correctness Correlation.** Let $Z_u = \mathbb{I}\{X_u = Y\}$, where $Y$ is the correct option. Denote single-trial accuracy as $p = \mathbb{P}(X_u = Y)$. Then for any pair $(Z_u, Z_v)$: Under the "copy" case: $\mathbb{P}(Z_u = Z_v = 1) = p$. Under the "independent" case: $\mathbb{P}(Z_u = Z_v = 1) = p^2$.

Therefore, the covariance is

$$\mathrm{Cov}(Z_u, Z_v) = \mathbb{E}[Z_u Z_v] - p^2 = \lambda(p - p^2) = \lambda p(1 - p),$$

and the correlation is

$$\rho(Z_u, Z_v) = \frac{\mathrm{Cov}(Z_u, Z_v)}{p(1 - p)} = \lambda. \tag{2}$$

Thus, the average correlation $\overline{\rho} = \lambda$.

### B.1.3 MAIN PROOF: MONOTONICITY OF MV IMPROVEMENT

Let $A_{\mathrm{MV}}(U; \lambda)$ be the expected MV accuracy under dependency level $\lambda$, and let $A_{\mathrm{single}} = p$ be the single-trial accuracy.

We decompose MV accuracy by conditioning on the latent sampling regime:

$$A_{\mathrm{MV}}(U; \lambda) = \lambda \cdot A_{\mathrm{MV}}(U; \mathrm{copy}) + (1 - \lambda) \cdot A_{\mathrm{MV}}(U; \mathrm{iid}). \tag{3}$$

In the "copy" case, all predictions are identical, so MV is equivalent to a single trial: $A_{\mathrm{MV}}(U; \mathrm{copy}) = p$. In the "iid" case, predictions are independent, and MV aggregates $U$ samples from $\pi$; here, accuracy improves with $U$, approaching 1 as $U \to \infty$ if $p > \frac{1}{K}$. Thus:

$$A_{\mathrm{MV}}(U; \lambda) = \lambda p + (1 - \lambda) A_{\mathrm{MV}}(U; 0), \tag{4}$$

$$\Delta A_{\mathrm{MV}}(U; \lambda) := A_{\mathrm{MV}}(U; \lambda) - p = (1 - \lambda)(A_{\mathrm{MV}}(U; 0) - p). \tag{5}$$

The improvement $\Delta A_{\mathrm{MV}}(U; \lambda)$ is thus a linear function decreasing in $\lambda$, and since $\lambda = \overline{\rho}$ (from equation 2) and $\overline{\mathrm{NMI}}$ increases with $\lambda$, MV improvement is monotonically decreasing in both.

### B.1.4 COROLLARY (EXTREMES)

If $\lambda = 1$ (i.e., $\overline{\rho} = 1$ or $\overline{\mathrm{NMI}} = 1$), then all predictions are identical and MV offers no improvement:

$$\Delta A_{\mathrm{MV}}(U) = 0.$$

If $\lambda = 0$ (i.e., predictions are independent) and $p > \frac{1}{K}$, then:

$$A_{\mathrm{MV}}(U) \to 1 \quad \text{as} \quad U \to \infty.$$

$\square$

### B.1.5 Discussion

This result formalizes an intuitive principle: confidence-based aggregation (e.g., MV) helps only when predictions are sufficiently diverse. High dependency, measured either via correctness correlation or mutual information, reduces the effective information gain from additional samples. Empirical results confirm this trend across VLMs and datasets: MV yields larger gains when dependency is low.

### B.2 Proof of Theorem 2

*Proof.* **Setup.** Fix a $K$-way classification item with true label $Y$. Let $u^\star := \arg\max_u p_u(Y)$ be the best model and define $c^\star := \Pr(\hat{y}_{u^\star} = Y)$. Let $\mathcal{B} = \{u \neq u^\star\}$ be the set of non-best models, with $|\mathcal{B}| \geq 2$.

**Coupling among non-best models.** Introduce a latent variable $L \in \{\text{copy}, \text{iid}\}$: - With probability $\lambda$, $L = \text{copy}$ and all non-best models predict a shared label $W$; define $\bar{c} := \Pr(W = Y)$. - With probability $1 - \lambda$, $L = \text{iid}$ and the non-best predictions are drawn independently.

**Step 1: Accuracy of ETTC.** Under Assumption 1, ETTC selects $\hat{y}_{u^\star}$, so:

$$A_{\min H} = \Pr(\hat{y}_{u^\star} = Y) = c^\star. \tag{6}$$

**Step 2: Accuracy of MV.** By law of total probability:

$$A_{\text{MV}}(\lambda) = \lambda \Pr(\widehat{Y}_{\text{MV}} = Y \mid L = \text{copy}) + (1 - \lambda) A_{\text{MV}}(0). \tag{7}$$

Under $L = \text{copy}$, all non-best models predict $W$, forming a majority:

$$\Pr(\widehat{Y}_{\text{MV}} = Y \mid L = \text{copy}) = \Pr(W = Y) = \bar{c}. \tag{8}$$

Plugging into equation 7, we recover:

$$A_{\text{MV}}(\lambda) = \lambda \bar{c} + (1 - \lambda) A_{\text{MV}}(0). \tag{9}$$

**Step 3: Difference and monotonicity.** Subtracting equation 9 from equation 6:

$$A_{\min H} - A_{\text{MV}}(\lambda) = \lambda(c^\star - \bar{c}) + (1 - \lambda)(c^\star - A_{\text{MV}}(0)). \tag{10}$$

This gap is nondecreasing in $\lambda$:

$$\frac{d}{d\lambda}(A_{\min H} - A_{\text{MV}}(\lambda)) = A_{\text{MV}}(0) - \bar{c} \geq 0.$$

**Step 4: Dominance threshold.** Let

$$\lambda^\star = \max\left\{0, \frac{A_{\text{MV}}(0) - c^\star}{A_{\text{MV}}(0) - \bar{c}}\right\}.$$

Then for all $\lambda \geq \lambda^\star$, ETTC outperforms MV; if $\bar{c} < c^\star$ and $\lambda > \lambda^\star$, the gap is strict. $\square$

**Remarks.** - Since $u^\star$ is the best model, typically $\bar{c} < c^\star$ unless all models perform equally well. - If $A_{\text{MV}}(0) \leq c^\star$, then $\lambda^\star = 0$: ETTC dominates MV at all dependency levels. - Under the copy-or-independent model, the average correctness correlation among non-best models equals $\lambda$ (see App. B.1), providing a direct link between dependency and the TTC advantage.

## C Experiment Settings

### C.1 Dataset

We evaluate our methods on six diverse multi-choice benchmarks spanning three domains: mathematical reasoning (MathVista, MathVision), diagram-based QA (TQA, ScienceQA), and general visual understanding (MMStar, MMMU). Tab. 3 summarizes key statistics, including dataset size, official split used, and number of answer options. Note that some datasets contain variable numbers of options (e.g., 2 - 9 in MMMU), which adds to the challenge and makes majority voting less stable. This diversity ensures our evaluation reflects a wide range of real-world reasoning settings.

Table 3: Dataset statistics and characteristics used in our evaluation. Each dataset is categorized by its domain (Math, Diagram, or General), the evaluation split used (e.g., test or validation), the number of multiple-choice questions (**Size**), and the number of answer options per question (**Option Num.**).

| Dataset | Domain | Type | Size | Option Num. |
|---|---|---|---|---|
| MathVista | Math | testmini | 540 | 2–8 |
| MathVision | Math | test | 1,532 | 5 |
| TQA | Diagram | test | 3,285 | 4 |
| ScienceQA | Diagram | test | 2,017 | 2–5 |
| MMStar | General | val | 1,500 | 4 |
| MMMU | General | val | 805 | 2–9 |

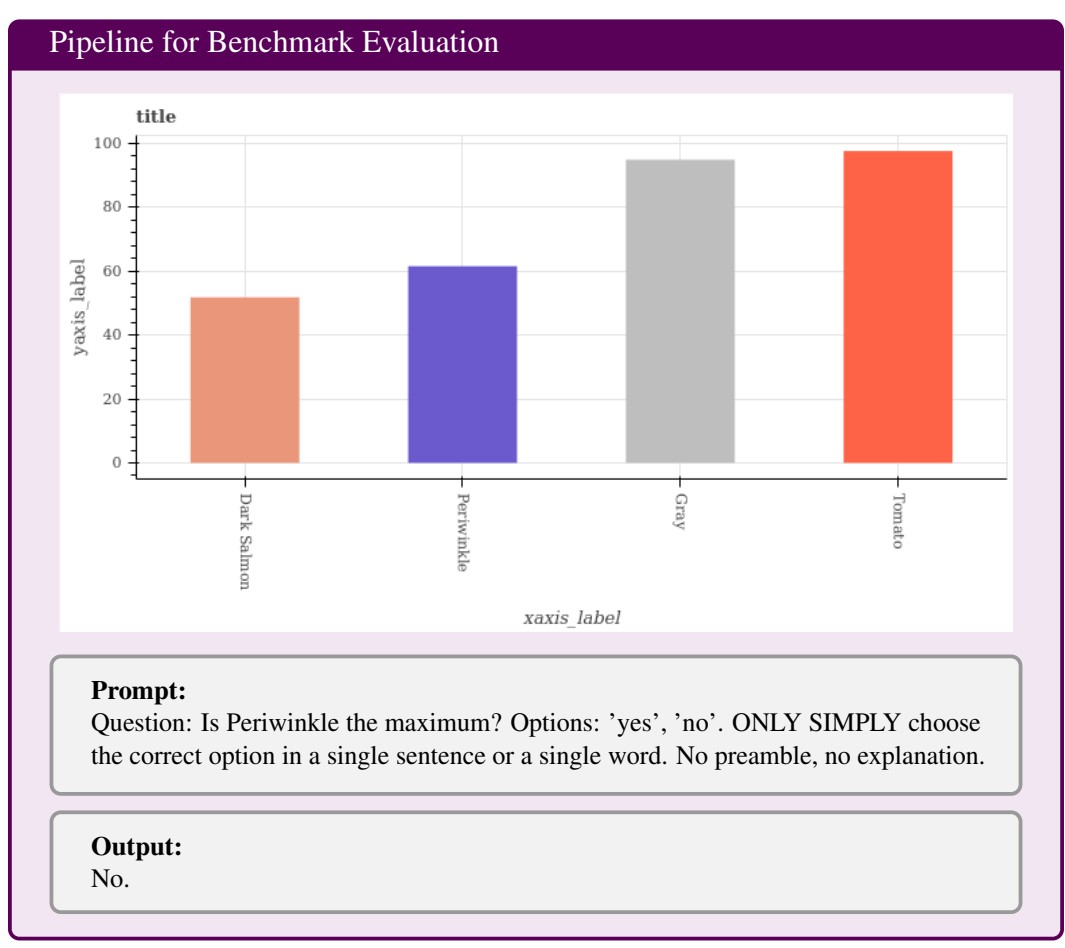

Figure 4: Example of a direct QA prompt used for evaluating model predictions without reasoning.

## C.2  PROMPT

To ensure consistency and minimize response variance across models, we standardize the prompting format in all benchmark evaluations. Specifically, we use a direct QA prompt without explanation, and a chain-of-thought (CoT) style prompt when evaluating reasoning performance or conducting consistency analysis. Below, we show two representative examples for comparison. The image and question are kept identical, while only the prompt template changes.

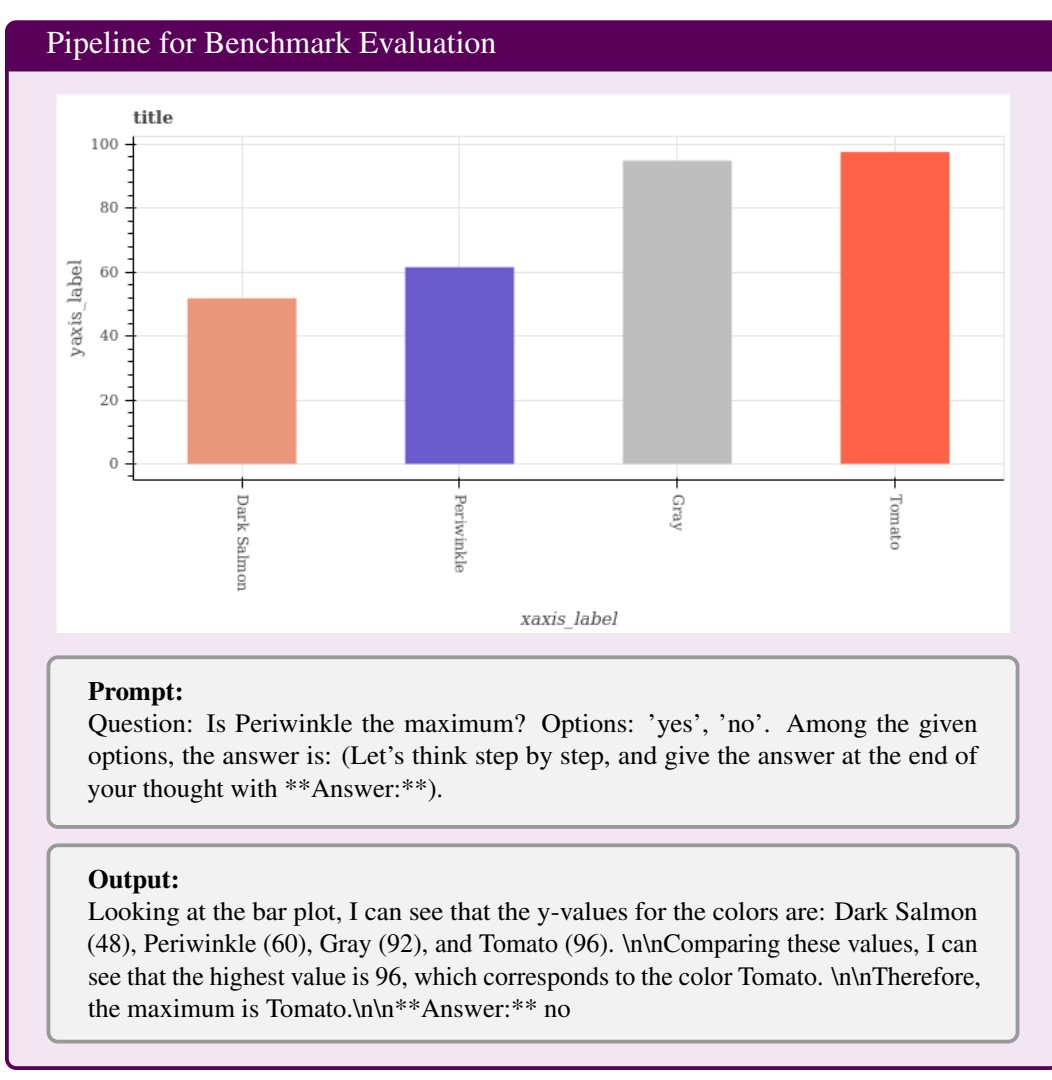

Figure 5: Example of a chain-of-thought (CoT) prompt used to elicit intermediate reasoning steps. This format is used when analyzing consistency or measuring correctness under step-by-step reasoning.

## C.3 BASELINES

To better assess the reliability of CoT responses, we include several shallow feature-based baselines. These models predict the correctness of a response using surface-level properties, without access to model internals or gradient signals.

Table 4: Pivot phrases categorized by reasoning function.

| Reasoning Type | Example Phrases |
|---|---|
| Realization | "wait", "oh", "actually", "I missed something" |
| Verification | "let me doublecheck", "to verify", "checking again" |
| Exploration | "what if", "another way to look at this", "alternatively" |
| Integration | "now I see how", "this connects back to", "putting this together" |

**Pivot words.** Pivot words are rhetorical expressions that signal shifts in reasoning, such as realization, verification, or synthesis. Prior work (Lippmann & Yang, 2025) suggests that the presence of such expressions often correlates with more deliberate and structured reasoning. We use a curated list of phrases categorized by rhetorical function, shown in Tab. 4. These are used as features for correctness prediction (e.g., counting their presence in CoTs).

Table 5: Vague expressions used in model reasoning, grouped by rhetorical effect.

| Reasoning Type | Example Phrases |
|---|---|
| **Uncertainty** | "maybe"", "possibly", "perhaps", "probably", "might be", "could be", "it seems" |
| **Hedging** | "somewhat", "rather", "kind of", "sort of", "generally", "typically" |

**Vague words.** Vague expressions are often used to hedge or express uncertainty, and may correlate with lower confidence or correctness in model reasoning. We group these into two categories, uncertainty and hedging—based on their rhetorical function. See Tab. 5.

Table 6: Overview of lexical and stylistic features used for CoT-based prediction.

| Feature | Modeling Method |
|---|---|
| **Token Number** | Measures the number of tokens in the CoT response. Longer responses may indicate more reasoning steps, though excessive length may signal loops or noise. We vectorize it as $1/\text{Token Number}$. |
| **Lexical Diversity** | Captures vocabulary richness by counting the number of unique tokens. Low diversity often suggests repetition. We vectorize it as $1/\text{Vocabulary Size}$. |
| **Pivot Word Number** | Counts the number of pivot expressions from Tab. 4, indicating structured reasoning or correction. We vectorize it as $1/\text{Pivot Word Number}$. |
| **Vague Word Number** | Counts the number of vague phrases from Tab. 5, which may reflect uncertainty or low confidence. We vectorize it as $1 - 1/\text{Vague Word Number}$. |

**Feature-All.** We also define a feature set that combines lexical and stylistic signals for each CoT response. Specifically, we consider four interpretable features: response length (token count), lexical diversity (unique token count), number of pivot words, and number of vague words. See Tab. 6 for detailed definitions. For prediction, we compute the sum of these feature values for each example, encouraging longer, more expressive, and more structured responses, while penalizing vague expressions. The model response with the highest total score is selected as the final prediction.

# D    SUPPLEMENTARY RESULTS

## D.1    MV IMPROVEMENT VS. $\overline{\text{NMI}}$ AND CORRELATION

While the overall trends in Figs. 6 and 7 are consistent with our theoretical expectations, MathVision stands out as an exception. Specifically, we observe weaker or even inverted correlation between prediction dependency and MV improvement on this dataset. A likely explanation is that MathVision poses significantly higher difficulty for current VLMs, its average accuracy across models is around 30%, which suggests that models are often uncertain or guessing. In such low-performance regimes, prediction behaviors may become erratic or overly stochastic, reducing the reliability of entropy, correlation, and voting-based signals. As a result, the dependency measures may not reflect meaningful error structure, making MV behavior less predictable.

## D.2    EMPIRICAL EVIDENCE TO SUPPORT ASSUMPTION

Fig. 8 shows the relationship between normalized entropy $\widetilde{H}_u$ and accuracy across multiple models on six benchmarks. We observe a strong inverse correlation between entropy and accuracy, consistent

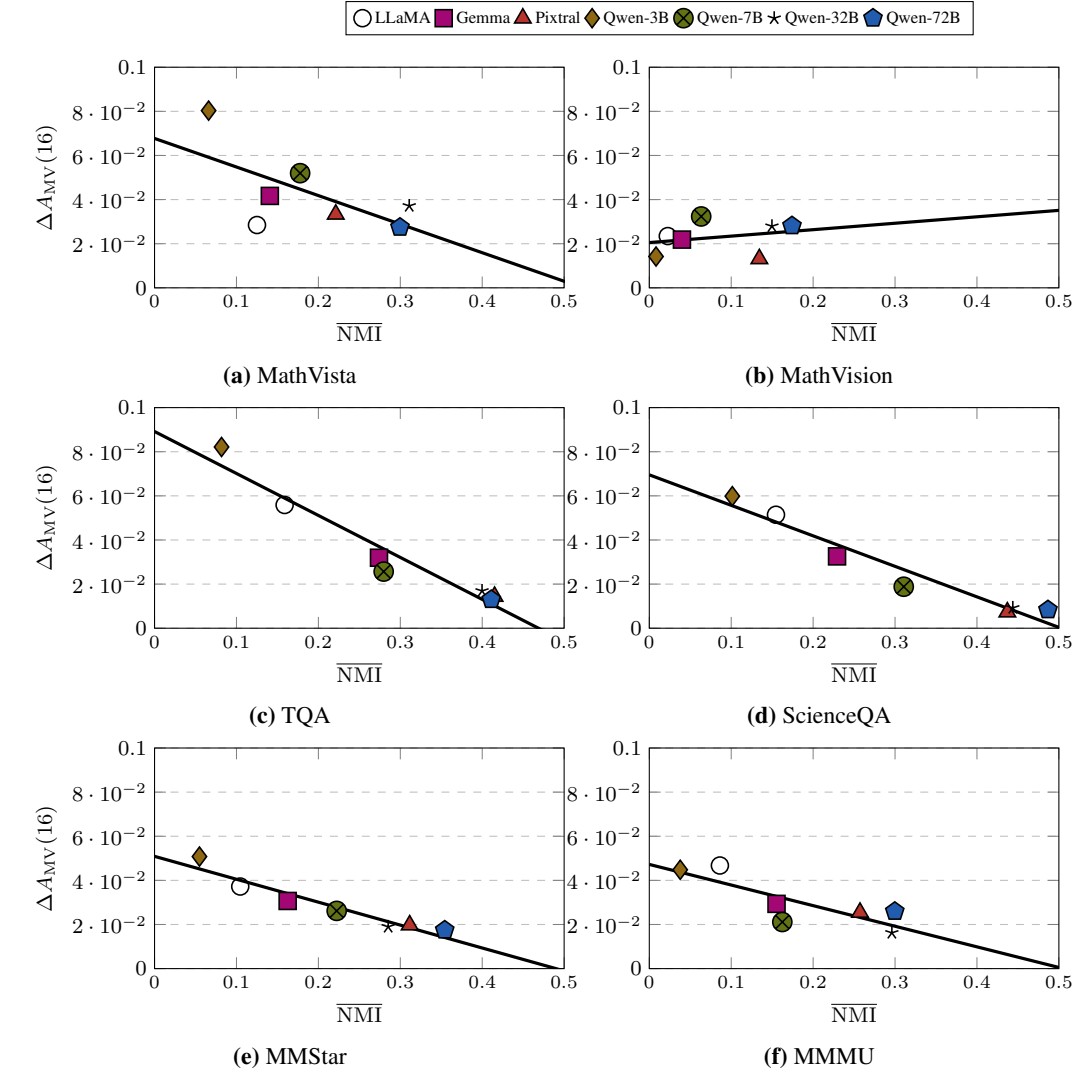

Figure 6: MV improvement $\Delta A_{\mathrm{MV}}(16)$ plotted against average pairwise normalized mutual information ($\overline{\mathrm{NMI}}$) for each model on each dataset. A negative trend suggests that higher prediction dependency reduces the benefit of majority voting.

with our Entropy-Accuracy Monotonicity assumption (Assumption 1). Higher-performing models generally exhibit lower entropy, indicating more confident and reliable predictions.

## D.3 SUPERVISED ETTC

We provide additional details on the supervised variant of ETTC, which learns from a small set of labeled question–model pairs when low entropy is a *reliable* signal of correctness.

**Problem setting.** Given $Q$ questions and $M$ models, each model $u$ produces a predictive distribution $p_{qu}(\cdot)$ over $K$ options for question $q$, aggregated over $U{=}16$ stochastic decoding samples (see § 4). The goal is to learn a function that predicts whether a model's low-entropy output is likely to be correct.

**Feature construction.** For each $(q, u)$ pair, we compute two features:

$$\widetilde{H}_{qu} := -\frac{1}{\log K} \sum_{k=1}^{K} p_{qu}(k) \log p_{qu}(k), \quad \mathrm{RelEnt}_{qu} := \frac{\widetilde{H}_{qu} - \min_v \widetilde{H}_{qv}}{\max_v \widetilde{H}_{qv} - \min_v \widetilde{H}_{qv}}.$$

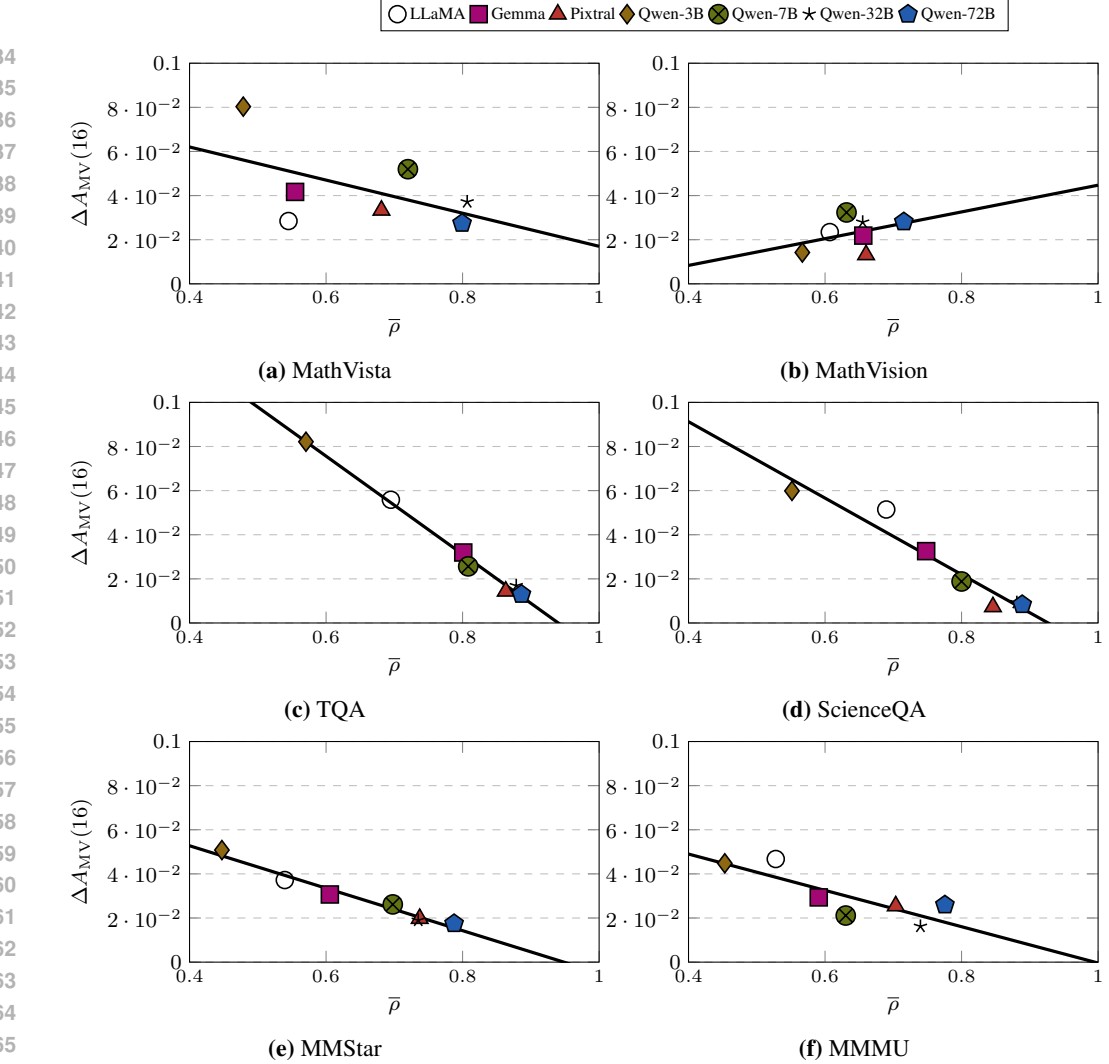

**Figure 7:** MV improvement $\Delta A_{\mathrm{MV}}(16)$ versus average pairwise accuracy correlation ($\overline{\rho}$). Consistent with theory, stronger dependency (i.e., higher $\overline{\rho}$) corresponds to smaller gains from majority voting.

Here $\widetilde{H}_{qu}$ is the normalized entropy of model $u$, while $\mathrm{RelEnt}_{qu}$ contextualizes this entropy relative to other models for the same question. The final feature vector is $(\widetilde{H}_{qu}, \mathrm{RelEnt}_{qu}) \in \mathbb{R}^2$.

**Labels and classifier.** The binary label is

$$Z_{qu} := \mathbb{I}\{\hat{y}_{qu} = Y_q\},$$

where $\hat{y}_{qu}$ is the top-1 prediction and $Y_q$ is the ground truth. We train a logistic regression classifier to predict $\Pr(Z_{qu} = 1)$ from the entropy features.

**Training protocol.** To simulate low-resource conditions, we use two-fold cross-validation across questions: each dataset is split into halves, one for training and one for testing, with roles reversed in a second run. This prevents test leakage and mimics scenarios where only limited annotations are available.

**Inference rule.** At test time, for each $(q, u)$ we compute the adjusted score

$$\mathrm{Score}_{qu} := \widetilde{H}_{qu} \cdot (1 - \hat{p}_{qu}),$$

where $\hat{p}_{qu}$ is the predicted correctness probability from the classifier. We then select the model with the lowest score:

$$u_q^\star := \arg\min_u \mathrm{Score}_{qu}, \quad \widehat{Y}_q := \hat{y}_{qu_q^\star}.$$

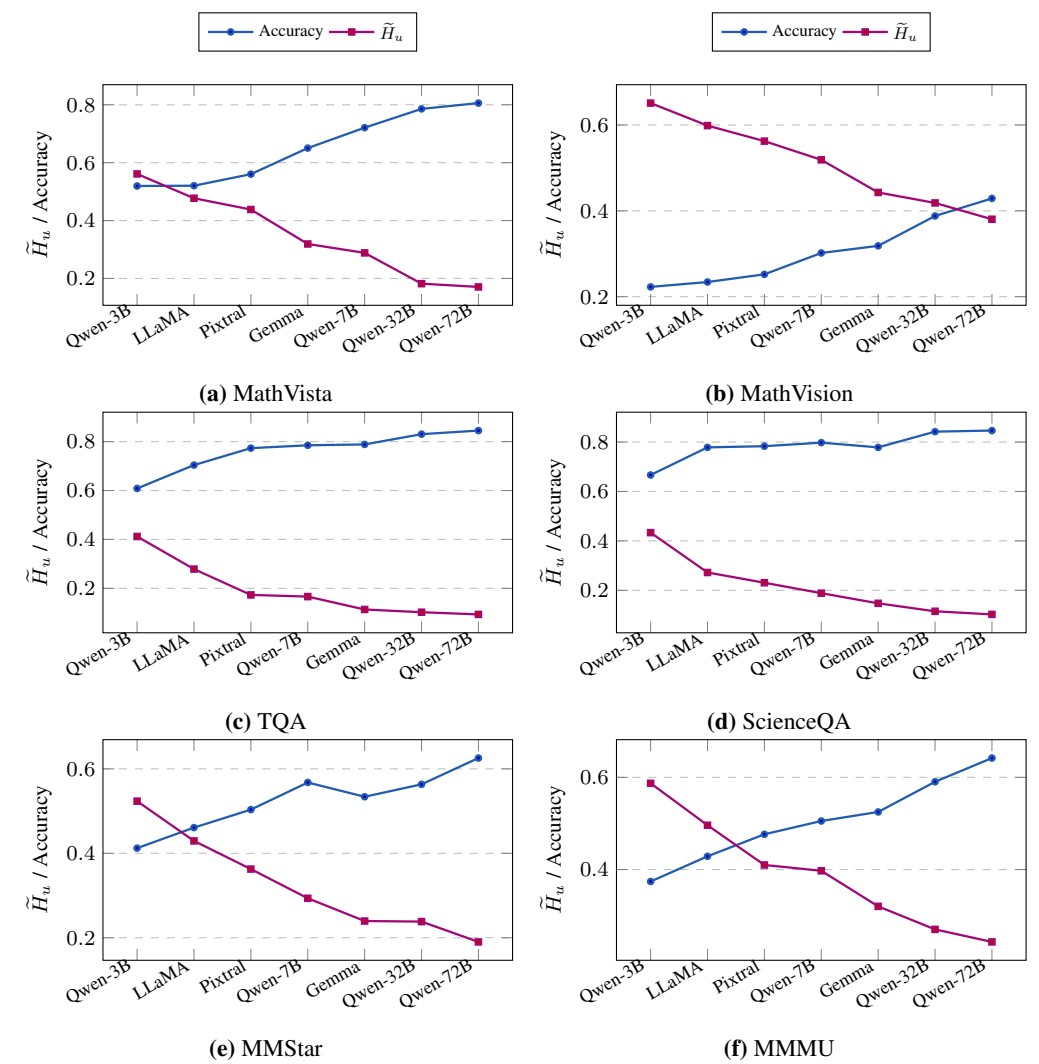

Figure 8: Correlation between normalized entropy $\widetilde{H}_u$ and accuracy across models on six benchmarks, supporting the Entropy–Accuracy Monotonicity assumption (Assumption 1).

This rule penalizes overconfident but unreliable predictions while rewarding trustworthy ones.

Table 7: Evaluation results across datasets for **Similar Size Models** and **Same Family Models**. Columns show the average single-model accuracy (Average), MV, (unsupervised) ETTC, and supervised variant of ETTC.

| Accuracy % | Similar Size Models | | | | Same Family Models | | | |
|---|---|---|---|---|---|---|---|---|
| | Avg. | MV | ETTC | Sup. ETTC$_\Delta$ | Avg. | MV | ETTC | Sup. ETTC$_\Delta$ |
| MathVista | 61.30 | 68.33 | 75.93 | $79.63_{3.70\uparrow}$ | 70.80 | 83.15 | 84.44 | $84.81_{0.37\uparrow}$ |
| MathVision | 27.66 | 32.05 | 35.57 | $36.62_{1.05\uparrow}$ | 33.53 | 41.32 | 44.84 | $46.34_{1.50\uparrow}$ |
| TQA | 76.28 | 83.65 | 83.90 | $84.14_{0.24\uparrow}$ | 76.73 | 84.90 | 86.70 | $86.70_{0.00\uparrow}$ |
| ScienceQA | 78.44 | 85.52 | 85.28 | $85.97_{0.69\uparrow}$ | 78.82 | 84.04 | 85.03 | $86.07_{1.04\uparrow}$ |
| MMStar | 51.65 | 59.27 | 60.07 | $60.67_{0.60\uparrow}$ | 54.22 | 61.00 | 63.73 | $65.07_{1.34\uparrow}$ |
| MMMU | 48.39 | 53.66 | 58.63 | $59.01_{0.38\uparrow}$ | 52.79 | 58.63 | 65.34 | $66.46_{1.12\uparrow}$ |
| **Average** | 57.29 | 63.75 | 66.56 | $67.67_{1.11\uparrow}$ | 61.15 | 68.84 | 71.68 | $72.58_{0.90\uparrow}$ |

**Results.** As shown in Tab. 7, supervised ETTC outperforms both MV and unsupervised ETTC across datasets and ensemble settings. Gains are largest on ambiguous tasks (e.g., MathVision, MMStar, MMMU), where entropy alone is less reliable. Even with only two-fold cross-fitting and no extra supervision, the classifier learns to identify failure modes of entropy selection, making more robust choices and underlining the value of combining entropy with supervised error modeling.

## LIMITATIONS

Our study focuses on multiple-choice visual reasoning tasks and assumes access to model confidence scores via output distributions. The proposed methods, especially entropy-based selection, may not directly generalize to open-ended tasks or models lacking probabilistic outputs. Additionally, while our evaluation covers diverse datasets and model ensembles, the gains of supervised entropy-based TTC depend on the quality and availability of annotated examples, which may be costly to obtain in some domains. Lastly, our analysis assumes that entropy correlates with accuracy, which may not hold for all models or tasks.

## LLM USAGE

We used ChatGPT as general-purpose assistive tools during the preparation of this paper. Specifically, LLMs were employed for polishing grammar, improving clarity, formatting LaTeX, generating illustrative figures, and debugging minor code snippets. LLMs were not involved in research ideation, experimental design, or the development of theoretical results.

