# OpenReview forum: "Diversity Matters: Revisiting Test-Time Compute in Vision-Language Models"
_ICLR.cc/2026/Conference — Submitted to ICLR 2026_

### Official Review · Reviewer_ZYoX · 2025-10-23

**Soundness:** 3
**Presentation:** 3
**Contribution:** 2
**Rating:** 4
**Confidence:** 5

**Summary:**

This paper systematically studies Test-Time Compute (TTC) strategies for visual reasoning with Vision-Language Models (VLMs), extending methods originally designed for LLMs. It finds that feature-based heuristics (e.g., reasoning length, pivot words) fail, and majority voting (MV) yields only modest gains due to limited prediction diversity. The authors provide a theoretical explanation linking MV's effectiveness to output dependency and introduce Entropy-based TTC (ETTC), which selects the most confident prediction via predictive entropy. ETTC generalizes MV, performs better under dependency, and is effective in multi-model ensembles, even allowing smaller models to enhance larger ones. Empirical results across seven VLMs and six benchmarks show that ETTC consistently outperforms both MV and the strongest single model without extra training.

**Strengths:**

1. The paper systematically examines TTC across multiple VLMs and benchmarks, bridging an important gap between LLM and VLM reasoning.
2. It provides a clear analytical link between prediction correlation and ensemble effectiveness, offering principled insight into why traditional voting fails.
3. ETTC is elegant, computationally light, and backward compatible with existing TTC setups—making it practical and generalizable.
4. The discovery that smaller models can enhance larger ones when weighted by confidence sounds interesting.

**Weaknesses:**

1. This work compares mainly to majority voting and other heuristics, but omits several recent, high-performing TTC methods that explicitly address confidence calibration and adaptive compute, such as:
Efficient Test-Time Scaling via Self-Calibration (Huang et al., 2025)
https://arxiv.org/pdf/2503.00031
COME: Test-Time Adaptation by Conservatively Minimizing Entropy (Zhang et al., 2025)
https://openreview.net/forum?id=506BjJ1ziZ
2. Why restrict evaluation to multimodal models (MLLMs/VLMs) rather than also testing on pure LLM reasoning benchmarks? Was there something specific about the vision-language setting, such as multimodal uncertainty, input grounding, or modality-induced diversity, that made TTC particularly interesting or challenging here?
3. The paper does not quantify inference overhead or efficiency trade-offs of the proposed method vs the baselines.

**Questions:**

1. Did you control for dataset bias or label imbalance that might affect entropy distributions or voting behavior?
2. How sensitive is ETTC to the number and composition of models in the ensemble? Does performance plateau or degrade beyond a certain number of weaker models?
3. In the same-family experiments (Qwen 3B–72B), what explains cases where smaller models overrode larger ones? Could you add some examples?

---

> ### Author Response · Authors · 2025-11-21
> **Response by the authors (Part 1)**
>
> We thank the reviewer for the constructive feedback and for highlighting the clarity, generality, and computational efficiency of ETTC, as well as the practical insight that smaller models can help larger ones. We address all concerns below.
>
> ---
>
> ***Point 1**: This work compares mainly to majority voting and other heuristics, but omits several recent, high-performing TTC methods...*
>
> > **Response**: We thank the reviewer for pointing out these relevant works. We will include both methods (Huang et al., 2025; COME, 2025) in the related work section. Importantly, both methods require a training stage or model adaptation, while our focus is strictly on pure inference-time strategies (e.g., Best-of-N / feature-base TTC and Self-Consistency / voting). When restricted to inference-only settings, these methods effectively reduce to feature-base / voting-like strategies, and they do not support multi-model ensembles. We will clarify this distinction more clearly in the revision.
>
>
> ---
>
> ***Point 2**: Why restrict evaluation to multimodal models (MLLMs/VLMs) rather than also testing on pure LLM reasoning benchmarks?*
>
> > **Response**: Our primary goal is to study TTC in visual reasoning, where perception bottlenecks and vision-language alignment introduce unique challenges absent in text-only settings. However, ETTC itself is general. As suggested, we have now evaluated ETTC on three thinking LLMs (Qwen3-Thinking 4B/30B/235B) on ARC (easy set, https://arxiv.org/abs/1803.05457) and MMUL-Pro (math set, https://arxiv.org/abs/2406.01574), and we observe consistent improvements over MV. This demonstrates that our findings extend beyond VLMs. We would include more results on LLMs / Thinking models in the next version.
> >
> > | Combination (ARC-Easy) | Min. | Max. | Avg. | MV | ETTC |
> > |---|---|---|---|---|---|
> > | 4B, 30B |  0.9599 | 0.9714 | 0.9656 | 0.9769 | 0.9878 |
> > | 4B, 235B | 0.9599  |  0.9772 | 0.9686  | 0.9769  |  0.9878 |
> > | 30B, 235B | 0.9714 | 0.9772 | 0.9743 | 0.9874 | 0.9895 |
> > | 4B, 30B, 235B | 0.9599  | 0.9772  |  0.9695 | 0.9891  | 0.9899  |
> >
> > | Combination (MMLU-Pro-Math) | Min. | Max. | Avg. | MV | ETTC |
> > |---|---|---|---|---|---|
> > | 4B, 30B | 0.8116 | 0.9412 | 0.8764 | 0.8934 | 0.9408 |
> > | 4B, 235B | 0.8116 | 0.9431 | 0.8773 | 0.8979 | 0.9467 |
> > | 30B, 235B | 0.9412 | 0.9431 | 0.9421 | 0.9504 | 0.9519 |
> > | 4B, 30B, 235B | 0.8116 | 0.9431 | 0.8986 | 0.9482 |0.9482  |
>
>
> ---
>
> ***Point 3**: The paper does not quantify inference overhead or efficiency trade-offs of the proposed method vs the baselines.*
>
> > **Response**: ETTC introduces negligible computational overhead. The only additional cost is computing entropy on the answer distribution, which is trivial compared to model inference. We will state this explicitly.
>
>
> ---
>
> ***Question 1**: Did you control for dataset bias or label imbalance that might affect entropy distributions or voting behavior?*
>
> > **Response**: We use standard benchmarks with well-balanced label distributions. Moreover, ETTC relies on normalized measures (e.g., entropy), which further mitigates potential effects of dataset imbalance. We find no evidence that bias affects entropy-based ordering in our experiments.
>
> ---
>
> ***Question 2**: How sensitive is ETTC to the number and composition of models in the ensemble?*
>
> > **Response**: Following this suggestion, we expanded our ablations to include a broader set of ensemble combinations (strong + weak models). Results on Qwen models across MathVista consistently show that ETTC yields positive gains under all tested configurations. We include these results in the revision.
> >
> >| Combination       | Min.    | Max.    | Avg.    | MV      | ETTC    |
> >|-----------|---------|---------|---------|---------|---------|
> >| 3B, 7B | 0.5194  | 0.7208  | 0.6201  | 0.6981  | 0.7926  |
> >| 3B, 32B | 0.5194  | 0.7858  | 0.6526  | 0.7204  | 0.8333  |
> >| 3B, 72B | 0.5194  | 0.8058  | 0.6626  | 0.7315  | 0.8481  |
> >| 7B, 32B | 0.7208  | 0.7858  | 0.7533  | 0.8148  | 0.8278  |
> >| 7B, 72B   | 0.7208  | 0.8058  | 0.7633  | 0.8222  | 0.8463  |
> >| 32B, 72B | 0.7858  | 0.8058  | 0.7958  | 0.8444  | 0.8426  |
> >| 3B, 7B, 32B | 0.5194  | 0.7858  | 0.6753  | 0.8130  | 0.8241  |
> >| 3B, 7B, 72B | 0.5194  | 0.8058  | 0.6820  | 0.8222  | 0.8444  |
> >| 3B, 32B, 72B | 0.5194  | 0.8058  | 0.7037  | 0.8370  | 0.8463  |
> >| 7B, 32B, 72B | 0.7208  | 0.8058  | 0.7708  | 0.8370  | 0.8426  |
>
>
> ---
>
> ***Question 3**: What explains cases where smaller models overrode larger ones?*
>
> > **Response**: Two factors contribute to this phenomenon. (1) Empirically, lower-entropy predictions correlate with correctness (Assumption 1; Appendix D.2). (2) Due to decoding randomness, larger models occasionally produce uncertain or incorrect answers, while smaller models may output correct and confident predictions (our theoretical discussions). ETTC leverages this by selecting the prediction with the lowest entropy, enabling smaller models to help correct occasional failures of larger ones.

---

> ### Author Response · Authors · 2025-11-21
> **Response by the authors (Part 2)**
>
> **Summary:** We thank the reviewer for the detailed feedback and for highlighting the practicality, efficiency, and clarity of our work. In addition to proposing ETTC, a lightweight and inference‑only TTC method, our paper provides a theoretical framework explaining the limitations of majority voting through prediction dependency, offering insight that complements and strengthens our empirical findings. We appreciate the reviewer's suggestions on additional baselines and ablations, and we are happy to answer any follow‑up questions.

---

### Official Review · Reviewer_eriH · 2025-10-29

**Soundness:** 2
**Presentation:** 4
**Contribution:** 2
**Rating:** 4
**Confidence:** 4

**Summary:**

- This work presents a systematic study of test-time compute for reasoning of vision-language models, evaluating seven VLMs using six benchmarks.
- The authors examine two paradigms: feature-based scoring of chain-of-thought (CoT) traces and confidence-based aggregation via majority voting (MV).
- The authors first studied TTC in a single-model setting and concluded that MV does not perform well there due to a lack of diversity (even with CoT prompting).
- The authors introduced Entropy-based Test-time Consistency (ETTC), which selects the prediction with the lowest entropy (i.e., most confident), exploiting confidence gaps between models.

**Strengths:**

- Multiple benchmarks were used to validate their TTC strategy, covering math reasoning, diagram understanding, and general visual reasoning.
- The authors presented their motivation and the background for this work well. The analysis of single-model MV shows only small gains, which motivates this work further.
- The authors verified ETTC with two ensemble configurations: 1) diverse models of different families, 2) scaled models developed by the same model developers. It's informative that the authors checked both cross-family diversity and same-family scaling.

**Weaknesses:**

- It would be nice to see results on frontier VLMs. This framework can be further validated with thinking models, but was mainly validated with non-frontier VLMs using just chain-of-thought prompting (i.e., the model is explicitly asked to think step-by-step). It's hard to say how much better MV or even their entropy-based approach with an ensemble of models will perform compared to just a single, actually top-tier frontier model alone, which should be confident in its answers when evaluated on these standard VLM benchmarks.
- The scope of this work is limited to or dedicated to VLMs. However, why is it limited to VLMs? More concretely, I would like more empirical evidence that backs the author's claim in the introduction and section 3, where this problem "...is further exacerbated in VLMs due to the perception bottleneck, visual content must first be interpreted before any meaningful variation can emerge." Please back this with experiments (e.g., text-only LLM MCQ using ETTC, and visual corruption evaluations where perception really matters).
- This work examines test-time compute for VLMs, which appears to be a general proposal to improve TTC for VLMs overall. However, results are only on MCQ task--does this framework hold for open-ended QA?

**Questions:**

- Should there be more ablations on the number of models with a mix of strong + weak models? It would be nice if the authors showed diminishing returns as the number of models increases and related gains to diversity and dependency measures. It's hard to fully trust the main results, given that ETTC was mainly run on two model ensemble configurations.
- Ensembling is a compute trade-off. Should there be more analysis of the trade-off between compute cost and accuracy, and between the TTC budget and accuracy?

---

> ### Author Response · Authors · 2025-11-21
> **Response by the authors**
>
> We thank the reviewer for the constructive feedback and for recognizing the clarity of our motivation and the comprehensiveness of our empirical evaluation. Below we address the points raised.
>
> ---
>
> ***Point 1 & Point 2**: This framework can be further validated with thinking models.
> The scope of this work is limited to or dedicated to VLMs. However, why is it limited to VLMs?*
>
> > **Response**: We appreciate this helpful suggestion. Our main focus is on VLMs because visual reasoning differs substantively from text reasoning: images present dense parallel information that must first be grounded, which limits diversity and makes TTC particularly challenging. That said, ETTC is modality-agnostic and can be applied directly to LLMs and thinking models.
> >
> > Following the reviewer’s suggestion, we extended ETTC to three thinking LLMs (Qwen3-Thinking-4B, 30B, 235B) on ARC (easy set, https://arxiv.org/abs/1803.05457) and MMUL-Pro (math set, https://arxiv.org/abs/2406.01574). The results (shown in the below table) demonstrate consistent gains over MV, confirming ETTC's generality beyond VLMs. We would include more results on LLMs / Thinking models in the next version.
> >
> > | Combination (ARC-Easy) | Min. | Max. | Avg. | MV | ETTC |
> > |---|---|---|---|---|---|
> > | 4B, 30B |  0.9599 | 0.9714 | 0.9656 | 0.9769 | 0.9878 |
> > | 4B, 235B | 0.9599  |  0.9772 | 0.9686  | 0.9769  |  0.9878 |
> > | 30B, 235B | 0.9714 | 0.9772 | 0.9743 | 0.9874 | 0.9895 |
> > | 4B, 30B, 235B | 0.9599  | 0.9772  |  0.9695 | 0.9891  | 0.9899  |
> >
> > | Combination (MMLU-Pro-Math) | Min. | Max. | Avg. | MV | ETTC |
> > |---|---|---|---|---|---|
> > | 4B, 30B | 0.8116 | 0.9412 | 0.8764 | 0.8934 | 0.9408 |
> > | 4B, 235B | 0.8116 | 0.9431 | 0.8773 | 0.8979 | 0.9467 |
> > | 30B, 235B | 0.9412 | 0.9431 | 0.9421 | 0.9504 | 0.9519 |
> > | 4B, 30B, 235B | 0.8116 | 0.9431 | 0.8986 | 0.9482 |0.9482  |
>
>
> ---
>
> ***Point 3**: However, results are only on MCQ task--does this framework hold for open-ended QA?*
>
> > **Response**: We focus on MCQ for controlled evaluation. However, prior work (e.g., Huang et al., 2025) shows that open-ended responses can be clustered into candidate sets, making them compatible with MCQ-style reasoning. ETTC immediately applies to these transformed settings, since it only requires an empirical answer distribution. We will clarify this in the revision.
>
> ---
>
> ***Question 1**: Should there be more ablations on the number of models with a mix of strong + weak models?*
>
> > **Response**: We appreciate this constructive suggestion. We extended our experiments to a broader set of ensemble combinations across both strong and weak models. The new results (Qwen models on MathVista) show consistent positive gains for ETTC across all combinations, further validating its robustness to diverse ensemble compositions.
> >
> >| Combination       | Min.    | Max.    | Avg.    | MV      | ETTC    |
> >|-------------------|---------|---------|---------|---------|---------|
> >| 3B, 7B            | 0.5194  | 0.7208  | 0.6201  | 0.6981  | 0.7926  |
> >| 3B, 32B           | 0.5194  | 0.7858  | 0.6526  | 0.7204  | 0.8333  |
> >| 3B, 72B           | 0.5194  | 0.8058  | 0.6626  | 0.7315  | 0.8481  |
> >| 7B, 32B           | 0.7208  | 0.7858  | 0.7533  | 0.8148  | 0.8278  |
> >| 7B, 72B           | 0.7208  | 0.8058  | 0.7633  | 0.8222  | 0.8463  |
> >| 32B, 72B          | 0.7858  | 0.8058  | 0.7958  | 0.8444  | 0.8426  |
> >| 3B, 7B, 32B       | 0.5194  | 0.7858  | 0.6753  | 0.8130  | 0.8241  |
> >| 3B, 7B, 72B       | 0.5194  | 0.8058  | 0.6820  | 0.8222  | 0.8444  |
> >| 3B, 32B, 72B      | 0.5194  | 0.8058  | 0.7037  | 0.8370  | 0.8463  |
> >| 7B, 32B, 72B      | 0.7208  | 0.8058  | 0.7708  | 0.8370  | 0.8426  |
>
> ---
>
> ***Question 2**: Should there be more analysis of the trade-off between compute cost and accuracy, and between the TTC budget and accuracy?*
>
> > **Response**: The compute-accuracy trade-off is dominated by the cost of generating $U$ outputs. ETTC adds only a negligible post-processing step (computing entropy), which is far cheaper than inference. Thus, ETTC achieves accuracy gains without materially increasing computational cost. We will clarify this in the revision.
>
> ---
>
> **Summary:** We thank the reviewer for the constructive comments and for acknowledging the clarity of our motivation and evaluation. Beyond ETTC itself, our work contributes a systematic theoretical and empirical characterization of TTC in VLMs, identifying perception bottlenecks and prediction dependency as key challenges that distinguish VLMs from text‑only models. We appreciate the reviewer's suggestions on extending experiments to thinking models and on ensemble ablations, which we have incorporated, and we are happy to answer any follow‑up questions.

---

### Official Review · Reviewer_LGSs · 2025-10-29

**Soundness:** 3
**Presentation:** 3
**Contribution:** 3
**Rating:** 6
**Confidence:** 3

**Summary:**

The paper studies the efficacy of the Test Time compute techniques in VLMs. The authors show that in the single model setting, feature based TTC methods fail to improve accuracy in VLMs, and confidence based majority voting yield 2-4% average gain. The main insight from the paper is that effectiveness of majority voting is monotonically decresing in prediction dependency and therefore the paper introduces Entropy based TTC, which consistently surpasses individual model and majority voting.

**Strengths:**

- The paper is well written and easy to follow.
- The paper introduces ETTC, which is a novel technique to ensemble aggregation of VLMs and it addresses the problem of correlated errors in majority voting. This enables smaller models to enhance larger models.
- The evaluation in the paper is comprehensive using VLMs such as Qwen, Llama, Gemma, Pixtra across various reasoning domains such as MathVista, TQA, MMMU.
- The theoretical analysis (Theorem 1) grounds the empirical observations about majority voting limitations.

**Weaknesses:**

- The key idea of ETTC relies on the assumption that predictive entropy can be used as a proxy for correctness. But poorly calibrated models can make this method less effective. Though authors mention this, I would encourage authors to analyze the robustness of their method under such scenarios.
- ETTC requires access to full predictive distribution of the VLMs which may not be accessible for black box models available under API calls.
- The paper focuses on the multiple choice visual question answering tasks and it is unclear how the proposed method would perform on the open ended generation tasks.

**Questions:**

- How would the performance gap between ETTC and Majority voting change if the VLMs are intentionally miscalibrated?
- What is the computational overhead of ETTC. Both ETTC and MV require generating U samples, how much cost is for the post processing step?
- Since the goal is compute efficiency, is the proposed method robust to weak small models?

---

> ### Author Response · Authors · 2025-11-21
> **Reponse by the authors**
>
> We sincerely thank the reviewer for the thoughtful feedback and for highlighting the novelty of our ETTC method, the strength of our theoretical and empirical analyses, and the breadth of our evaluation across models and datasets. We address each point below.
>
> ---
>
> ***Point 1 & Question 1**: ETTC relies on the assumption that predictive entropy can be used as a proxy for correctness. But poorly calibrated models can make this method less effective.
> How would the performance gap between ETTC and Majority voting change if the VLMs are intentionally miscalibrated?*
>
> > **Response**: We appreciate this insightful point. ETTC is grounded in the empirical observation that correct predictions consistently exhibit lower-entropy answer distributions across all models and datasets we studied (Appendix D.2). This makes entropy a practical and informative confidence indicator in typical VLM settings.
> >
> > We agree that under intentionally miscalibrated or adversarial conditions this correlation may weaken. To handle such cases, we introduce a supervised extension of ETTC (Appendix D.3), which learns model-specific coefficients based on accuracy signals. This variant improves robustness when some models are miscalibrated and further supports the reviewer's concern.
>
> ---
>
> ***Point 2**: ETTC requires access to full predictive distribution of the VLMs which may not be accessible for black box models available under API calls.*
>
> > **Response**: ETTC does not require internal logits or access to hidden states. It only relies on the empirical distribution of final answer options, which can be obtained from multiple sampled outputs in black-box API settings. Thus, ETTC is fully compatible with API-based VLMs.
>
>
> ---
>
> ***Point 3**: The paper focuses on the multiple choice visual question answering tasks and it is unclear how the proposed method would perform on the open ended generation tasks.*
>
> > **Response**: We use multiple-choice QA because it offers a controlled and widely adopted evaluation protocol for visual reasoning. For open-ended formats, prior work (e.g., Huang et al., 2025) shows that responses can be reliably clustered or grouped into candidate sets. ETTC directly applies to such derived option spaces, since it only operates on the empirical answer distribution. We will clarify this in the revision.
>
> ---
>
> ***Question 2**: What is the computational overhead of ETTC. Both ETTC and MV require generating U samples, how much cost is for the post processing step?*
>
> > **Response**: ETTC adds negligible overhead beyond generating $U$ samples. The only additional operation is computing the entropy of a $K$-way distribution, which takes microseconds in Python and is several orders of magnitude cheaper than a single model forward pass. Thus, ETTC introduces effectively no extra compute burden in TTC scenarios.
>
>
> ---
>
> ***Question 3**: Since the goal is compute efficiency, is the proposed method robust to weak small models?*
>
> > **Response**: Yes. As shown in the same-family experiments (e.g., Qwen-VL-3B + 7B + 32B + 72B), smaller models still provide informative low-entropy predictions. In several cases, adding weaker models further improves the performance of the strongest model (Qwen-VL-72B), offering direct evidence that ETTC is robust to heterogeneous ensembles. As requested by other reviewers, we add a more comprehensive ensemble test, and results show that ETTC could improve large model with small ones across all settings (Qwen models on MathVista).
> >
> >| Combination       | Min.    | Max.    | Avg.    | MV      | ETTC    |
> >|-------------------|---------|---------|---------|---------|---------|
> >| 3B, 7B  | 0.5194  | 0.7208  | 0.6201  | 0.6981  | 0.7926  |
> >| 3B, 32B  | 0.5194  | 0.7858  | 0.6526  | 0.7204  | 0.8333  |
> >| 3B, 72B  | 0.5194  | 0.8058  | 0.6626  | 0.7315  | 0.8481  |
> >| 7B, 32B | 0.7208  | 0.7858  | 0.7533  | 0.8148  | 0.8278  |
> >| 7B, 72B     | 0.7208  | 0.8058  | 0.7633  | 0.8222  | 0.8463  |
> >| 32B, 72B    | 0.7858  | 0.8058  | 0.7958  | 0.8444  | 0.8426  |
> >| 3B, 7B, 32B       | 0.5194  | 0.7858  | 0.6753  | 0.8130  | 0.8241  |
> >| 3B, 7B, 72B       | 0.5194  | 0.8058  | 0.6820  | 0.8222  | 0.8444  |
> >| 3B, 32B, 72B      | 0.5194  | 0.8058  | 0.7037  | 0.8370  | 0.8463  |
> >| 7B, 32B, 72B      | 0.7208  | 0.8058  | 0.7708  | 0.8370  | 0.8426  |
>
> ---
>
> **Summary**: We thank the reviewer for the thoughtful evaluation and for recognizing the novelty of ETTC, the strength of our theoretical analysis, and the breadth of our empirical results. We would like to emphasize that our core contribution is not limited to proposing a new TTC method, but also to providing a principled understanding of why traditional voting strategies are fundamentally constrained by prediction dependency in VLMs: a theoretical insight that guides and motivates ETTC. We appreciate the reviewer's constructive suggestions on calibration, ensemble composition, and practical considerations, and we are happy to answer any follow‑up questions.

---

### Author Response · Authors · 2025-11-27
**Gentle Reminder for Discussion**

Dear Reviewers,

Thank you again for the thoughtful and constructive feedback. We’ve carefully addressed each of your comments in our rebuttal, including additional experiment results (e.g., extended ensemble settings and LLM evaluations) that we hope directly speak to your concerns.

As the discussion period concludes soon, we would greatly appreciate it if you could take a moment to review our responses. We would also be happy to clarify anything further or follow up on any remaining questions you may have.

Thank you again for your time and consideration.

Best,

The Authors

---

### Author Response · Authors · 2025-11-30
**Rebuttal Summary for AC**

Dear AC,

We would like to briefly summarize the key concerns raised by the reviewers and how we addressed them in our rebuttal.


---

***1. Ensemble setting and robustness (Reviewer LGSs, eriH, ZYoX).***

> Reviewers requested broader ensemble configurations and sensitivity analyses. We added new experiments with diverse combinations of strong and weak models, and ETTC consistently outperforms majority voting across all tested settings.
>
> *See: Response to LGSs — Question 3; eriH — Question 1; Response to ZYoX — Question 2.*

---

***2. Generalization beyond VLMs (Reviewer eriH, ZYoX).***

> To confirm that our theoretical insights are not specific to VLMs, we extended our evaluation to three thinking LLMs (Qwen3-Thinking 4B/30B/235B) on ARC and MMLU-Pro. ETTC again shows consistent gains, demonstrating that our analysis and method generalize to text-only reasoning tasks.
>
> *See: Response to eriH — Points 1 & 2; Response to ZYoX — Point 2.*


---

***3. Robustness to miscalibration (Reviewer LGSs).***

> The reviewer asked whether ETTC remains effective when models are intentionally miscalibrated. We clarified the underlying assumption and introduced a supervised variant of ETTC (Appendix D.3) that improves robustness in such cases.
>
> *See: Response to LGSs — Point 1 & Question 1.*


---

***4. Additional baselines and compute overhead (Reviewer LGSs, eriH, ZYoX).***

> We clarified that our work focuses strictly on inference-only pattern. and confirmed that ETTC adds negligible computational overhead beyond standard TTC sampling.
>
> *See: Response to LGSs — Question 2; eriH — Question 2; ZYoX — Points 1 & 3.*


---

***5. Applicability to open-ended QA and API settings (Reviewer LGSs, eriH).***

> We clarified that ETTC only requires empirical answer distributions from sampled outputs (not logits), making it fully compatible with black-box API models. Prior work shows that open-ended QA can be reliably converted to MCQ-style candidate sets, to which ETTC applies directly.
>
> *See: Response to LGSs — Points 2 & 3; Response to eriH — Point 3.*

---

All remaining minor points (dataset bias, formatting, etc.) were addressed in the rebuttal and will be reflected in the final version.

Overall, we believe the additional experimental results and clarifications directly resolve the main concerns raised by the reviewers, particularly regarding ensemble behavior and generalization beyond VLMs. We hope this summary is helpful for AC to make the decision.

Best,

The Authors

---

### Meta-Review · Area_Chair_ZG6P · 2026-01-06

**Summary:**

The paper systematically investigates test-time compute strategies for Vision-Language Models, proposing an Entropy-based TTC (ETTC) method to improve ensemble aggregation by leveraging prediction confidence. During the review process, the reviewers acknowledged the clarity of the presentation and the theoretical insights linking prediction diversity to the limitations of majority voting. However, significant concerns were raised regarding the novelty of the approach and the scope of the claims. Specifically, reviewers noted that the reliance on predictive entropy is a standard heuristic in uncertainty estimation, and the exclusion of recent state-of-the-art test-time scaling baselines limited the assessment of the method's true impact. Furthermore, while the authors provided a strong rebuttal that extended experiments to "thinking" LLMs and clarified the inference-only setting, the consensus remains that the work offers incremental technical advancements over established ensemble techniques. Given the concerns about the marginal nature of the contribution relative to the current literature, the paper is not recommended for acceptance at this time.

**Reviewer Concerns:**

Addressed concerns are (1) generalization beyond VLMs: In response to Reviewers eriH and ZYoX, the authors successfully provided additional results on "thinking" LLMs (Qwen-Thinking on ARC and MMLU-Pro), demonstrating that the method is not strictly limited to the visual modality. (2) Ensemble Robustness: The concerns regarding the interaction between weak and strong models (Reviewer LGSs, eriH) were addressed through new ablation studies, showing that the entropy-based selection remains robust even when weaker models are included in the ensemble. (3) Applicability & Overhead: The authors clarified that the method is compatible with black-box APIs (as it relies on output distributions rather than internal logits) and entails negligible computational overhead.

Outstanding concerns are (1) Limited Novelty: While the application to VLM ensembles is empirically validated, the core technical contribution—using predictive entropy as a confidence proxy to weight or select predictions—is a well-established heuristic in uncertainty estimation. The reviewers noted that this approach, while effective, represents a somewhat incremental technical advance over standard majority voting. (2) Baselines and Scope: Reviewer ZYoX pointed out the lack of comparison against recent, more sophisticated Test-Time Compute (TTC) methods (e.g., self-calibration or adaptation-based approaches). While the authors argued that their focus is strictly "inference-only" to exclude these baselines, this narrow scoping limits the paper's impact. By restricting the comparison primarily to Majority Voting and simple heuristics, the paper does not fully contextualize its performance against the state-of-the-art in adaptive compute.

**Reviewer Scores:**

Reviewer LGSs: 6 (Unchanged)
Reviewer eriH: 5 (Marginally Increased from 4)
Reviewer ZYoX: 4 (Unchanged)

---

### Decision · Program_Chairs · 2026-01-26

Reject